**Investigation**

# Hidden structure in polygenic scores and the challenge of disentangling ancestry interactions in admixed populations

Alan J. Aw (ID) ,[1,]* Ravi Mandla,[1] Zhuozheng Shi,[1] Penn Medicine Biobank[2,†] Bogdan Pasaniuc,[1] Iain Mathieson (ID) [1,]*

[1]Department of Genetics, Perelman School of Medicine, University of Pennsylvania, Philadelphia, PA 19104, United States
[2]Perelman School of Medicine, University of Pennsylvania, Philadelphia, PA 19104, United States

*Corresponding author: Alan J. Aw, Department of Genetics, Perelman School of Medicine, University of Pennsylvania, Philadelphia, PA 19104, United States. Email: alan.aw@pennmedicine.upenn.edu; Iain Mathieson, Department of Genetics, Perelman School of Medicine, University of Pennsylvania, Philadelphia, PA 19104, United States. Email: mathi@pennmedicine.upenn.edu
[†]List of authors provided in Supplemental Material.

The extent to which genetic effects vary across ancestries remains a central question in human genetics, with direct implications for improving polygenic prediction. Recent studies have found that average causal effects are similar across ancestries, but empirical evidence and theoretical considerations suggest that interactions may still contribute to poor portability of polygenic scores. Here, we leverage the ancestry mosaicism of admixed individuals to develop models of genetic interactions with local and global ancestry. We show that these models capture epistasis but are also consistent with highly similar average causal effects, clarifying that the latter observation does not exclude the possibility of gene–ancestry interaction. Next, we investigate how models of local and global interactions differ in polygenic prediction. Focusing on continuous traits, we show that if causal variants are known and causal effects differ across ancestries, then the global and local models can be differentiated using partial polygenic scores—scores computed on ancestry-specific genomic segments—while standard polygenic scores remain indistinguishable under both models. However, if causal variants are unknown, differences in linkage disequilibrium (LD) patterns confound this analysis, and the models are not practically distinguishable without knowledge of the LD between causal and tagging variants. Our findings motivate future developments of model-sensitive strategies for individual-level genetic risk prediction.

Keywords: admixed population; polygenic scores; gene-by-ancestry; epistasis; statistical genetics

## Introduction

A fundamental open question in human genetics is the extent to which the genetic basis of human diseases and traits is shared across individuals. For example, it is unclear how much low cross-population polygenic score portability can be attributed to differences in genetic effects across ancestries, or to what extent gene–environment and gene–gene interactions impact human disease. Hence, understanding the role of ancestry in variability of causal effect sizes has tremendous implications for understanding the genetic basis of disease and portability of genetic risk scores.

The standard approach to estimating similarity of causal effects involves cross-population analyses in which average population-level effect sizes estimated by large-scale genome-wide association studies (GWASs) are compared across different populations or across different ancestral segments within recently admixed populations (Fig. 1a). Such studies have generally found that population-level effects are similar across different groupings of individuals: an emerging consensus in the field is that causal effects are highly correlated across ancestries (Hou et al. 2023; Verma et al. 2024; Hu et al. 2025) and that differences in the phenotypic variance explained by tagging variants can be largely explained by differences in linkage disequilibrium and allele frequency (Wang et al. 2020).

As an alternative to studying population-level effects that average the individual-level effects across all individuals in the study, the distribution of individual-level effects within the population can provide insight into the impact of epistatic and gene–environment interactions. Unfortunately, estimating individual-level effects directly is impossible. However, both theoretical considerations and empirical analyses suggest that at least in some cases, gene–gene and gene–environment interactions should play a role in between-population effect heterogeneity, and thus potentially also in the heterogeneity of effects between individuals within a population (Kerin and Marchini 2020; Mostafavi et al. 2020; Mathieson 2021; Currant et al. 2023; Mackay and Anholt 2024; Veller et al. 2024).

Here, we leverage the unique feature of recently admixed genomes, which are mosaics of ancestrally varying segments leading to substantial variation in local ancestry between individuals within the same population. For example, African American genomes are composed of segments of African and European ancestries inherited within the past 15 generations. Many admixed populations, including African Americans, also exhibit substantial variation in global ancestry proportions. This variability in ancestry at both local and genome-wide levels allows us to develop statistical models of individual-level effects, where genetic and possibly environmental interactions are captured by ancestry

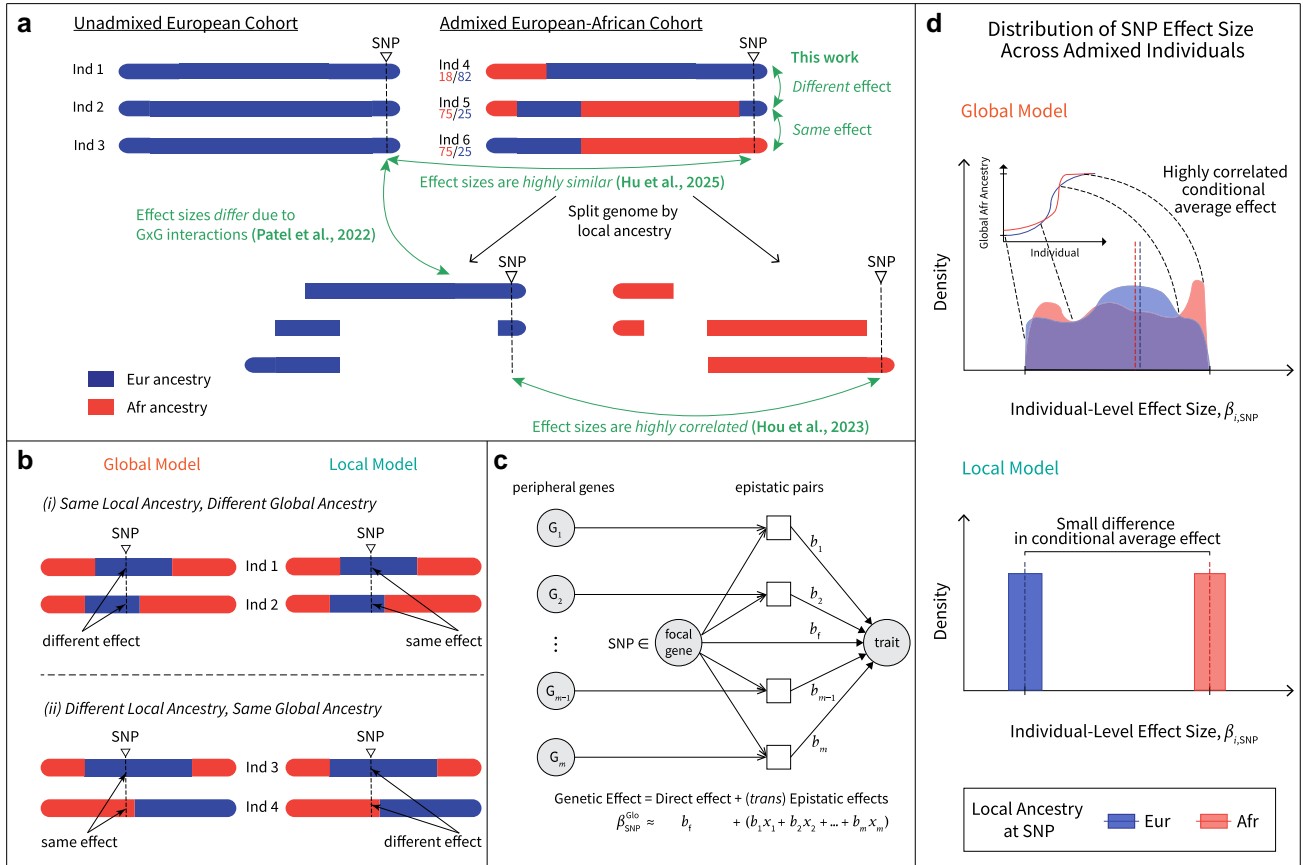

**Fig. 1.** a) Comparison of various approaches investigating variant effect heterogeneity by ancestry. Green arrows depict contexts between which allelic effects are compared. Individuals 4, 5 and 6 have their global African and European ancestries printed below their IDs. For standing height, a highly polygenic trait, several studies have reported quantities justifying high causal effect similarity. Chen et al. (2023) estimated high "trans-ethnic" genetic correlations $r_g \geq 0.93$, although Shi et al. (2021) found depletion of such correlations in functional regions. Hou et al. (2023) estimated high correlation of causal variant effects by local ancestry ($r_{admix} = 0.94$, albeit significantly less than 1), while Hu et al. (2025) reported high similarity of causal variant effects (95% CI of slope ratio $\rho$ contains 1). Height was not investigated in Patel et al. (2022). b) Two haplotypes of each of two pairs of individuals (1 and 2, 3 and 4) are illustrated, with the local and global models assigning same or different genetic effects at a focal variant ("SNP"). c) Illustration of a structural causal model of non-specific epistasis (Domingo et al. 2019) that the global model approximates. d) Visualization of distributions of individual effect sizes at a particular SNP, according to either the global model (Eq. (7)) or the local model (Eq. (6)). Dashed coloured lines depict average causal effects conditioned on local ancestry (i.e. "conditional average effect") as defined in Individual and Average Effects. Under the global model, between-individual variation in global African ancestry (top graph inset) leads to a spread of individual-level effect sizes at the SNP. As we verify in our analysis (Global model implies continuous distribution of individual level effect sizes, but still predicts highly similar average causal effects) and also show here through the closeness of the dashed coloured lines, this is nonetheless consistent with highly correlated average causal effect conditioned on local ancestry. The local model encodes the model assumptions of standard GWAS, where an average marginal effect applies to all individuals, and this effect may differ by ancestry given differences in allele frequency and linkage disequilibrium, or effect modifications due to *cis* interactions.

interactions. Hence, we introduce models that encode global and local ancestry contributions to complex traits, which capture *trans* and *cis* interaction effects respectively.

We investigate the implication of these models for average causal effects and polygenic score performance, using phased genotypes from admixed individuals of African and European ancestry in the Penn Medicine Biobank (PMBB). First, we show analytically and through simulations that high average causal effect similarity is consistent with a model of heterogeneous individual-level causal effects that vary continuously with global ancestry proportions. Moreover, in some cases it can also be consistent with a model in which causal effects vary discretely with local ancestry. Next, we show that when causal variants and their effects are known, these models can be differentiated by comparing total and partial polygenic scores. However, in the more realistic case where only tagging variants are available, the models cannot be distinguished because the difference

between the two models is confounded with LD differences between the populations.

Despite this limitation, our findings highlight the fact that highly similar average causal effects across local ancestries do not exclude the possibility of statistical gene–ancestry interactions. These findings motivate the need for methods that disentangle the contributions of local and global ancestries to inter-individual genetic effect heterogeneity, to enhance the precision of genetic risk predictions.

The remainder of the paper is organized as follows. In Materials and Methods, we introduce our quantitative models incorporating local and global ancestry interactions, and describe simulations we conduct using PMBB genotypes. In Results we report key findings about average causal effects and polygenic prediction. Finally, in Discussion we share the implications of our findings and some limitations of our approach. Formal arguments for our models and claims are provided in the Supplemental Material.

## Materials and methods

### Quantitative model

Let $n$ individuals genotyped across $p$ markers make up an admixed cohort. Individuals arise from $k$-way admixture; in this work $k = 2$ (African and European ancestries). Markers correspond to tagging variants included in a polygenic score. Assume that the genotypes are phased and their ancestries are known, so that we have a $n \times 2p$ matrix $\mathbf{X} = [\mathbf{X}^{(1)} \mid \mathbf{X}^{(2)}] = [x_{ij}^{(1)} \mid x_{ij}^{(2)}]$ denoting the individual-by-marker stacked haplotype matrix, and another $n \times 2p$ matrix $\mathbf{A} = [\mathbf{A}^{(1)} \mid \mathbf{A}^{(2)}] = [a_{ij}^{(1)} \mid a_{ij}^{(2)}]$ recording the local ancestries of each individual's two haplotypes. Note that $a_{ij}^{(h)} = 1$ if the ancestry inferred for haplotype $h$ at marker $j$ in individual $i$ is African, otherwise the inferred ancestry is European and $a_{ij}^{(h)} = 0$.

We also study the causal variants that the markers tag, so we have another $n \times 2p$ matrix $\mathbf{X}' = [\mathbf{X}'^{(1)} \mid \mathbf{X}'^{(2)}] = [x_{ij}'^{(1)} \mid x_{ij}'^{(2)}]$, where variant $j$ in $\mathbf{X}'$ is the causal variant tagged by variant $j$ in $\mathbf{X}$ defined in the previous paragraph. (Throughout this work, we differentiate causal from tagging quantities with a prime symbol, $'$.) Similarly, the local ancestry matrix for the causal variants is $\mathbf{A}' = [\mathbf{A}'^{(1)} \mid \mathbf{A}'^{(2)}] = [a_{ij}'^{(1)} \mid a_{ij}'^{(2)}]$. In addition to these matrices, we have linkage disequilibrium (LD) vectors that record LD between the $j$th causal and tagging variant. These are ancestry-specific and we denote them by $\boldsymbol{\lambda}^{\mathrm{Afr}} = (\lambda_j^{\mathrm{Afr}} : j = 1, \ldots, p)$ and $\boldsymbol{\lambda}^{\mathrm{Eur}} = (\lambda_j^{\mathrm{Eur}} : j = 1, \ldots, p)$.

We follow an approach to modeling phenotypes, whereby an individual $i$'s phenotype $y_i$ is of the form $y_i = \sum_{j=1}^{p} \left( \beta_{ij}^{(1)} x_{ij}'^{(1)} + \beta_{ij}^{(2)} x_{ij}'^{(2)} \right) + \varepsilon_i$. To specify the genetic effect $\beta_{ij}^{(h)}$ and to see how they depend on the haplotype $h$, we begin by defining two models, which differ in their assignment of genetic effects in admixed individuals based on ancestry. First, let ancestry-specific causal allele frequencies be denoted by $(f_j'^{\mathrm{Eur}} : j = 1, \ldots, p)$ and $(f_j'^{\mathrm{Afr}} : j = 1, \ldots, p)$. We compute the demeaned haplotypes

$$\hat{x}_{ij}'^{(h)} = \left( x_{ij}'^{(h)} - f_j'^{\mathrm{Afr}} \right) a_{ij}'^{(h)} + \left( x_{ij}'^{(h)} - f_j'^{\mathrm{Eur}} \right) \left( 1 - a_{ij}'^{(h)} \right) \qquad (h = 1, 2) \quad (1)$$

We demean haplotypes to ensure the cumulative genetic effects are invariant to labeling of alleles, since otherwise polygenic scores computed on non-demeaned haplotypes of admixed individuals shift differently depending on the local ancestry and polarization of the effect allele at the locus (see Supplemental Material S1 for details). Our approach is similar to Hu et al. (2025), who showed (see their Supplementary Note 2) that failing to demean haplotypes also leads to biased estimates of ancestry-specific effect sizes.

We consider variant effects for each single-ancestry group, $\beta_j'^{\mathrm{Eur}}$ (European) and $\beta_j'^{\mathrm{Afr}}$ (African); these quantities measure the average causal effect across all individuals homogeneous for that ancestry. Similar to Wang et al. (2020), we assume that the variant effects are jointly normally distributed.

$$\begin{bmatrix} \beta_j'^{\mathrm{Eur}} \\ \beta_j'^{\mathrm{Afr}} \end{bmatrix} \sim N\left( \begin{bmatrix} 0 \\ 0 \end{bmatrix}, \begin{bmatrix} \sigma_{\mathrm{Eur}}'^2 & \tau' \\ \tau' & \sigma_{\mathrm{Afr}}'^2 \end{bmatrix} \right), \quad (2)$$

where $\sigma_{\mathrm{Eur}}'^2$, $\sigma_{\mathrm{Afr}}'^2$ and $\tau'$ are defined by

$$\sigma_{\mathrm{Eur}}'^2 = \frac{r^2}{2 \sum_{j=1}^{p} f_j'^{\mathrm{Eur}} (1 - f_j'^{\mathrm{Eur}})} \quad (3)$$

$$\sigma_{\mathrm{Afr}}'^2 = \frac{r^2}{2 \sum_{j=1}^{p} f_j'^{\mathrm{Afr}} (1 - f_j'^{\mathrm{Afr}})} \quad (4)$$

$$\tau' = \frac{r^2 \rho}{2 \sqrt{\sum_{j=1}^{p} f_j'^{\mathrm{Eur}} (1 - f_j'^{\mathrm{Eur}})} \sqrt{\sum_{j=1}^{p} f_j'^{\mathrm{Afr}} (1 - f_j'^{\mathrm{Afr}})}} \quad (5)$$

and $r^2$ and $\rho$ are parameters that control the variance explained by genetic effects and the cross-ancestry correlation of causal effects. In particular, when $\rho = 1$ the causal effects are directly proportional to one another: collecting the causal effects for each ancestry into the vectors $\boldsymbol{\beta}'^{\mathrm{Afr}} = (\beta_j'^{\mathrm{Afr}} : j = 1, \ldots, p)$ and $\boldsymbol{\beta}'^{\mathrm{Eur}} = (\beta_j'^{\mathrm{Eur}} : j = 1, \ldots, p)$ we have $\boldsymbol{\beta}'^{\mathrm{Afr}} \propto \boldsymbol{\beta}'^{\mathrm{Eur}}$. This implies that they are both perfectly correlated (i.e. $\mathrm{cor}(\boldsymbol{\beta}'^{\mathrm{Afr}}, \boldsymbol{\beta}'^{\mathrm{Eur}}) = 1$) and perfectly similar (cosine similarity of $\boldsymbol{\beta}'^{\mathrm{Afr}}$ and $\boldsymbol{\beta}'^{\mathrm{Afr}}$ is 1). Equally important, however, they need not be identical.

Given ancestry-specific variant effects (Eq. (2)), we define the two models (Fig. 1b). The first model, called the *local model*, assumes that the effect size at a marker $j$ for an individual $i$ depends only on the local ancestry at the marker. Concretely, for haplotype $h$,

$$\beta_{ij}'^{\mathrm{Loc},h} = \beta_j'^{\mathrm{Eur}} \left( 1 - a_{ij}'^{(h)} \right) + \beta_j'^{\mathrm{Afr}} a_{ij}'^{(h)}. \quad (6)$$

Equation (6) models how *cis* epistasis induces causal effect heterogeneity (Aschard et al. 2015; see also Supplemental Material S2 for a rigorous argument). For the second model, called the *global model*, the effect size depends on the marker's global ancestry:

$$\beta_{ij}'^{\mathrm{Glo},h} = \beta_j'^{\mathrm{Eur}} \left( 1 - \overline{a_{i.}} \right) + \beta_j'^{\mathrm{Afr}} \overline{a_{i.}}, \quad (7)$$

where $\overline{a_{i.}}$ denotes the global African ancestry of individual $i$. The global model captures non-specific *trans* epistasis under a scenario where the set of *trans*-modifying variants contributing multiplicative epistatic effects is unknown (Fig. 1c illustrates a structural causal model). Specifically, the right hand side of Eq. (7) is approximately the expected effect size under this generative model, where the expectation is taken over the joint distribution of local ancestries and allelic dosages at the *trans*-modifying loci and focal variant (see Supplemental Material S2 for a rigorous argument). The global model may also capture G×E interactions (Park et al. 2018). We compute $\overline{a_{i.}}$ from data by averaging local ancestry across the individual's genome (see Supplemental Material S7 for details). This calculation was previously verified in Zaidi et al. (2023) to agree well with global ancestry estimated from ADMIXTURE (Alexander et al. 2009), and we further verify that they are empirically stable to leaving one chromosome out in our study cohort (Supplementary Fig. S1). Note that even though the effect size under the global model is indexed by haplotype to help with streamlining definitions of downstream quantities, it does not actually depend on the haplotype index (i.e. $\beta_{ij}'^{\mathrm{Glo},1} = \beta_{ij}'^{\mathrm{Glo},2}$). For each of these models, the phenotype is defined as

$$y_i = \sum_{j=1}^{p} \left( \beta_{ij}'^{,1} \hat{x}_{ij}'^{(1)} + \beta_{ij}'^{,2} \hat{x}_{ij}'^{(2)} \right) + \varepsilon_i, \quad (8)$$

where $\in \{\mathrm{Loc}, \mathrm{Glo}\}$) and $\varepsilon_i$ is environmental noise independent of genotypes, ancestry and genetic effects, and is drawn from a normal distribution such that the phenotype has approximately unit variance.

### Individual and average effects

We study genetic effects, including both causal and tagging effects. We make a distinction between individual and average causal effects, which clarifies what the local and global models imply

about both the allelic substitution effect for an admixed individual and the mean effect across individuals sharing a particular local ancestry—the latter of which was recently found to be largely invariant (i.e. "highly similar" or "highly correlated") to the particular ancestry (Hou et al. 2023; Hu et al. 2025).

Individual causal effects refer to the effect of a specific individual $i$ carrying a copy of the alternate allele, as shown in Eqs. (6) and (7). Average causal effects, however, measure the mean effect of carrying the same allelic copy across a specific group of individuals. In this paper, individuals are grouped by local ancestry at a specific locus, so that average causal effects for that locus can be compared across local ancestries by computing conditional averages (the conditioning variable being marker local ancestry). Concretely, these quantities are calculated by averaging Eqs. (6) and (7) across all individuals carrying one local ancestry at locus $j$, followed by all individuals carrying the other local ancestry. For example, under the global model the average causal effect for all individuals with African local ancestry is $\beta'_{\cdot j}|_{LA=Afr} := [n(\overline{a'^{(1)}_{\cdot j}} + \overline{a'^{(2)}_{\cdot j}})]^{-1}(\sum_{i:a'^{(1)}_{ij}=1} \beta'^{Glo,1}_{ij} + \sum_{i:a'^{(2)}_{ij}=1} \beta'^{Glo,2}_{ij})$. The corresponding quantity for individuals with European local ancestry is $\beta'_{\cdot j}|_{LA=Eur} := [n(2 - \overline{a'^{(1)}_{\cdot j}} - \overline{a'^{(2)}_{\cdot j}})]^{-1}(\sum_{i:a'^{(1)}_{ij}=0} \beta'^{Glo,1}_{ij} + \sum_{i:a'^{(2)}_{ij}=0} \beta'^{Glo,2}_{ij})$. Under the local model, the average causal effects are simply the ancestry-specific causal effects, $\beta'^{Afr}_j$ and $\beta'^{Eur}_j$. As Fig. 1d shows, one result of this work is that average causal effects can be highly similar, even if substantial inter-individual effect size heterogeneity exists.

Table 1 summarizes the various types of causal effects discussed in this work. Individual and average tagging effects are defined in a similar manner, with the latter averaged across all individuals sharing a particular local ancestry at the tagging locus.

## Polygenic scores

Polygenic scores are typically calculated from tagging variants and their effect sizes. Let ancestry-specific tagging allele frequencies be denoted by $(f^{Eur}_j : j = 1, \ldots, p)$ and $(f^{Afr}_j : j = 1, \ldots, p)$. Given causal and tagging allele frequencies, LD and causal effect sizes, the tagging effect sizes are as follows (Vukcevic et al. 2011; see also Supplemental Material S3 for a proof).

$$\beta^{Eur}_j = \beta'^{Eur}_j \lambda^{Eur}_j \sqrt{\frac{f'^{Eur}_j(1 - f'^{Eur}_j)}{f^{Eur}_j(1 - f^{Eur}_j)}} \tag{9}$$

$$\beta^{Afr}_j = \beta'^{Afr}_j \lambda^{Afr}_j \sqrt{\frac{f'^{Afr}_j(1 - f'^{Afr}_j)}{f^{Afr}_j(1 - f^{Afr}_j)}} \tag{10}$$

Here $\lambda_j$ is the ancestry-specific LD, or "correlation of frequencies," as described in Eq. (1) of Wright (1933). If we denote the scaling factors by $\theta^{Eur}_j = \lambda^{Eur}_j \sqrt{f'^{Eur}_j(1 - f'^{Eur}_j)}/\sqrt{f^{Eur}_j(1 - f^{Eur}_j)}$ and $\theta^{Afr}_j = \lambda^{Afr}_j \sqrt{f'^{Afr}_j(1 - f'^{Afr}_j)}/\sqrt{f^{Afr}_j(1 - f^{Afr}_j)}$, then using Eq. (2) we may express the distribution of tagging effects directly:

$$\begin{bmatrix} \beta^{Eur}_j \\ \beta^{Afr}_j \end{bmatrix} \sim N\left( \begin{bmatrix} 0 \\ 0 \end{bmatrix}, \begin{bmatrix} \sigma'^2_{Eur}(\theta^{Eur}_j)^2 & \tau'\theta^{Eur}_j\theta^{Afr}_j \\ \tau'\theta^{Eur}_j\theta^{Afr}_j & \sigma'^2_{Afr}(\theta^{Afr}_j)^2 \end{bmatrix} \right). \tag{11}$$

We assume in our analysis that tagging effects are estimated without bias, though controlling for confounding during estimation is an appreciably challenging task (see Veller and Coop 2024

and citations therein). A special case is when the tagging and causal variants coincide (such as in the unlikely scenario where one is able to identify all causal variants from statistical fine-mapping or experimental means); this corresponds to $\lambda^{Eur}_j = \lambda^{Afr}_j = 1$, $f'^{Eur}_j = f^{Eur}_j$ and $f'^{Afr}_j = f^{Afr}_j$. Consequently, the scaling factors $\theta^{Eur}_j = \theta^{Afr}_j = 1$ and so tagging effects are identical to causal effects. Another special case is when the two ancestral populations share the same LD and allele frequencies across tagging and causal variants. This corresponds to $\theta^{Eur}_j = \theta^{Afr}_j = \theta_j$ for all $j$, and so Eq. (11) reduces to Eq. (2) by having $\sigma'^2_{Eur}$, $\sigma'^2_{Afr}$ and $\tau'$ replaced by the scaled versions $\sigma'^2_{Eur}\theta^2_j$, $\sigma'^2_{Afr}\theta^2_j$ and $\tau'\theta^2_j$.

Given local ancestry, haplotype matrices and tagging effect sizes, we may compute polygenic scores. For an admixed individual $i$, let their (tagging variant) haplotype and local ancestry vectors be $\mathbf{x}_i = (x^{(1)}_{i1}, \ldots, x^{(1)}_{ip}, x^{(2)}_{i1}, \ldots, x^{(2)}_{ip})$ and $\mathbf{a}_i = (a^{(1)}_{i1}, \ldots, a^{(1)}_{ip}, a^{(2)}_{i1}, \ldots, a^{(2)}_{ip})$. Similar to our treatment of causal variants (Eq. (1)), we compute their demeaned haplotype dosages $\hat{x}^{(h)}_{ij}$ using tagging allele frequencies. Their *total polygenic score* is

$$\text{TotPGS}(\mathbf{x}_i, \mathbf{a}_i) = \sum_{j=1}^{p} \beta^{Eur}_j \left( \hat{x}^{(1)}_{ij} + \hat{x}^{(2)}_{ij} \right), \tag{12}$$

which is obtained by assigning European-ancestry tagging effect sizes (Eq. (9)) to each effect allele at marker $j$ regardless of the allele's local ancestry. Their *partial polygenic score* (see Bitarello and Mathieson 2020; Marnetto et al. 2020; Sun et al. 2024) on the other hand, is obtained by assigning effect sizes only to effect alleles at haplotype locus $j$ when the local ancestry is European (i.e. whenever $a^{(h)}_{ij} = 0$).

$$\text{ParPGS}(\mathbf{x}_i, \mathbf{a}_i) = \sum_{j=1}^{p} \beta^{Eur}_j \left[ \left(1 - a^{(1)}_{ij}\right)\hat{x}^{(1)}_{ij} + \left(1 - a^{(2)}_{ij}\right)\hat{x}^{(2)}_{ij} \right] \tag{13}$$

In Results we state our main findings on how partial and total polygenic scores behave under the local and global models. Throughout this work, we evaluate polygenic score behavior by computing the squared correlation between the polygenic score and the phenotype.

## Data description
### Penn Medicine Biobank

The PMBB (2020 release) consists of patients of the University of Pennsylvania Health System, with written consent provided to collect and store biological specimens and carry out DNA extraction and sequencing (Verma et al. 2022). Access to and analysis of data were approved by the Institutional Review Board at the University of Pennsylvania. We analyzed two groups of unrelated individuals: $n = 9{,}324$ individuals with mixed African and European ancestry and $n = 29{,}410$ individuals with broad European ancestry, hereafter called PMBB ADM and PMBB EUR cohorts. We use genotype array data comprising $p = 579{,}988$ biallelic variants for both groups.

### Genotype phasing and ancestry inference

We used local ancestry calls generated in Zaidi et al. (2023). Briefly, these were generated by phasing genotypes using Beagle (v5.4, Browning et al. 2021), before using RFMix (Maples et al. 2013) to infer local ancestry (setting $k = 2$) with genotypes from the 1000 Genomes Project (of CEU and YRI individuals) as reference. Local ancestry assignment was obtained by binarizing

**Table 1.** Genetic effect quantities and their interpretation.

| Locus-specific quantity | What it measures | Relevant samples | Ancestral background used for explicit comparisons | Examples |
|---|---|---|---|---|
| Average causal effect | Genetic effect of carrying one copy of an allele averaged across a group of individuals; quantity is group-specific and thus comparisons are between groups. | Two or more ancestrally homogeneous populations | Two populations of different homogeneous global ancestries (e.g. East Asians vs. non-admixed Europeans); implicitly excludes admixed individuals by categorizing global ancestries. | Shi et al. (2021) and Chen et al. (2023) |
| | | One admixed population | Two groups with different local ancestries at the locus (e.g. individuals with African local ancestry vs. individuals with European local ancestry). | Hou et al. (2023) and Hu et al. (2025) |
| Individual causal effect | Genetic effect of carrying one copy of an allele for one particular individual; individual-specific and not directly estimable. | One admixed population | Among individuals, their respective local and global ancestries (former is locus-specific; latter is not). | This work |
| | | Individuals within the same family | Ancestry is not presently considered. | Veller et al. (2024) |

posterior probabilities, and global ancestry for each individual was obtained as the overall proportion of inferred African (or European) ancestry.

### Allele frequency and LD calculation

We computed the following empirical allele frequency and LD quantities:

- *Local ancestry-conditioned allele frequencies*. Relevant only to PMBB ADM cohort, these include (1) the causal variants selected for the 50 sets of independent SNPs in Simulation Study 1, and (2) the causal and tagging variants selected using summary statistics for the six phenotypes in Simulation Study 2. We use these sample-specific quantities—$\hat{f}_j'^{\text{Eur}}$, $\hat{f}_j'^{\text{Afr}}$ and $\hat{f}_j^{\text{Eur}}$, $\hat{f}_j^{\text{Afr}}$—as empirical estimates of the quantities $f_j'^{\text{Eur}}, f_j'^{\text{Afr}}$ and $f_j^{\text{Eur}}, f_j^{\text{Afr}}$ as they appear in Quantitative Model (for example, in computing quantities in Eqs. (3)–(5) and demeaning haplotypes, but see next point).
- *European ancestry allele frequencies*. The causal and tagging variant allele frequencies computed on PMBB EUR cohort. We only use these quantities as empirical estimates of $f_j'^{\text{Eur}}, f_j^{\text{Eur}}$ when computing European tagging effect sizes, as described in Eq. (9). We do so to be consistent with the usage of PMBB EUR cohort to estimate European ancestry LD.
- *Linkage Disequilibrium (LD)*. These include African ancestry LD, estimated from the formula $\lambda = [\mathbb{P}(X = 1, X' = 1) - f \cdot f'] / \sqrt{f(1-f)f'(1-f')}$ (Wright 1933). Here $\mathbb{P}(X = 1, X' = 1)$ is computed on pairs of causal and tagging haplotypes sharing African local ancestry, and $f, f'$ are computed as local ancestry-conditioned allele frequencies. Because European local ancestry proportions are small resulting in noisy estimates of the preceding quantities, we instead used the PMBB EUR cohort to estimate European ancestry LD, where we computed the Pearson correlation of the unphased causal and tagging genotypes. We use these quantities, $\hat{\lambda}_j^{\text{Eur}}$ and $\hat{\lambda}_j^{\text{Afr}}$, as estimates for $\lambda_j^{\text{Eur}}, \lambda_j^{\text{Afr}}$ in computing African and European tagging effect sizes as described in Eqs. (9) and (10). This calculation is not relevant to Simulation Study 1.

### GWAS summary statistics and causal-tagging variant assignment

We obtained GWAS summary statistics for European ancestry individuals from the Pan UK Biobank Consortium (Karczewski et al. 2025). We considered six traits: Standing Height, Weight, Body Mass Index, Triglycerides, Neutrophil Count and Platelet Count. We performed four steps on these summary statistics files prior to running downstream analyses. First, we removed variants without available summary statistics (i.e. had NA values for $\beta$ and $s.e.(\beta)$); about 82% of variants were kept per trait. Next, we ran LIFTOVER (Hinrichs et al. 2006) to convert variant coordinates from hg19 to hg38 and enable variant matching between summary statistics and PMBB genotype markers; 99.97% of variants were lifted over successfully per trait. Third, we randomly removed duplicate variants, defined as multiple variants that share the same chromosome and position; less than 0.2% of variants were removed per trait. Finally, we converted log-scaled $p$-values ($-\log_{10}(P)$) to original $p$-values ($P$) for clumping and thresholding, by performing $x \mapsto 10^{-x}$ and taking the maximum of that value and $10^{-323}$ as the output to avoid rounding down to zero due to arithmetic underflow.

With processed summary statistics in hand, we constructed putatively causal and tagging variants for each trait. Briefly (details in Supplemental Material S7) we trained clumping and thresholding polygenic scores on the PMBB EUR cohort to estimate the median number of variants included in a PGS. Taking the top GWAS hits for each trait based on the median number of variants estimated, pairs of variants in high LD ($\lambda^2 \geq 0.8$) were assigned as putatively causal and tagging, with causal variant assigned to the GWAS hit and tagging variant assigned as the other variant. Filters were performed to ensure that tagging variants were not also appearing as a top GWAS hit. The number of causal-tagging variant pairs for each trait is summarized in Supplementary Table S2.

## Simulation overview

We perform two simulation studies using PMBB genotypes, one assuming causal variants and their effect sizes are known and another assuming polygenic scores can only be computed on tagging variants.

## Simulation study 1

We simulated phenotypes from PMBB ADM cohort genotypes using Eq. (8). To illustrate the relationship between ancestry and polygenic score portability, we split the ADM cohort into four subgroups based on global ancestry (mean global ancestries in subgroups are 0.61, 0.79, 0.85 and 0.92; see Genotype Phasing and Ancestry Inference for details on global ancestry calculation). To obtain approximately independent markers, we intersected LD blocks identified by LDᴇᴛᴇᴄᴛ (Berisa and Pickrell 2016) for 1000 Genomes European and African populations, before randomly selecting one marker from each autosomal block and evaluating independence using ꜰʟɪɴᴛʏ (Aw et al. 2024; see Supplemental Material S7 and Fig. S18). We performed the random selection 50 times to obtain 50 seeds, which we used for all simulations. (Each seed has $n =$ 9,324 individuals split into four groups, and $p = 1,536$ approximately independent markers.) We first simulated phenotypes under both global and local models (Eq. (8)), where for each seed 100 draws from each model were performed before computing squared correlations between phenotype and polygenic scores. Mean squared correlations were computed and 95% confidence intervals constructed. Altogether, we simulated $50 \times 100 \times 6 \times 6 \times 2 = 360,000$ phenotypes for the global and local models.

## Simulation study 2

We simulate traits using GWAS summary statistics for six phenotypes and real phased admixed genotypes. The phenotypes include standing height, weight, body mass index, triglycerides, neutrophil count and platelet count. We use GWAS summary statistics to avoid modeling the GWAS discovery process, and to ensure that our simulations reflect properties of real polygenic scores (e.g. realistic joint frequency distribution of causal and tagging variants). We consider six phenotypes to reflect a reasonable range of well-studied quantitative traits, so that any finding that holds across all of them is more likely generalizable. Causal variants are selected based on top GWAS hits, with tagging variants subsequently picked based on having a sufficiently large correlation with the causal variant in the PMBB EUR cohort (Supplemental Material S7). With causal and tagging variants in hand, we proceeded to simulate phenotypes under the global and local models, varying parameters $\rho$, $r^2$ and ensuring that $r^2$ ranges across a grid of 10 values representative of their empirical estimates (Supplemental Material S7). For each phenotype, $\rho$ ranges across 25 values spanning 0.2 and 1. Per choice of $(\rho, r^2)$, 200 draws from each model were performed. Altogether, we simulated $6 \times 25 \times 10 \times 200 \times 2 = 600,000$ phenotypes for the global and local models.

# Results
## Global model implies continuous distribution of individual level effect sizes, but still predicts highly similar average causal effects

We investigate implications of the global and local models for individual and average causal effects. On an individual level, the local model implies that any two individuals sharing the same local ancestry carry the same allelic substitution effect. The global model, however, stipulates that any two individuals sharing the same global ancestry carry the same allelic substitution effect. Because global ancestry is continuous while local ancestry is binary, the global model effectively implies a distribution of effect sizes across all admixed individuals (Fig. 1d).

To obtain average causal effects by local ancestry (see Individual and Average Effects), we derive analytical expressions for the distribution of average causal genetic effect by each local ancestry. (The distribution of average *tagging* effects by each local ancestry is described in Supplemental Material S8, Box D.)

**Proposition 1**(Joint Distribution of Local Ancestry Average Causal Effects). Under the local model, the joint distribution of average causal effects is the same as the original joint distribution of causal effects in the base model, Eq. (2). Let $\beta'_{.j}|_{\text{LA=Afr}}$ and $\beta'_{.j}|_{\text{LA=Eur}}$ denote the African and European local ancestry average causal effects under the global model. The joint distribution of these quantities is

$$\begin{bmatrix} \beta'_{.j}|_{\text{LA=Afr}} \\ \beta'_{.j}|_{\text{LA=Eur}} \end{bmatrix} \sim N\left( \begin{bmatrix} 0 \\ 0 \end{bmatrix}, \begin{bmatrix} u'_j & w'_j \\ w'_j & v'_j \end{bmatrix} \right),$$

where

$$u'_j = \sigma'^2_{\text{Eur}}\omega'^2_{1,j} + 2\tau'\omega'_{1,j}\omega'_{2,j} + \sigma'^2_{\text{Afr}}\omega'^2_{2,j}$$
$$v'_j = \sigma'^2_{\text{Eur}}\omega'^2_{3,j} + 2\tau'\omega'_{3,j}\omega'_{4,j} + \sigma'^2_{\text{Afr}}\omega'^2_{4,j}$$
$$w'_j = \sigma'^2_{\text{Eur}}\omega'_{1,j}\omega'_{3,j} + \tau'(\omega'_{2,j}\omega'_{3,j} + \omega'_{1,j}\omega'_{4,j}) + \sigma'^2_{\text{Afr}}\omega'_{2,j}\omega'_{4,j}$$

are terms in the covariance matrix, with quantities $\sigma'^2_{\text{Afr}}$, $\sigma'^2_{\text{Eur}}$ and $\tau'$ defined in Eqs. (3)–(5), and quantities $\omega'_{1,j}$, $\omega'_{2,j}$, $\omega'_{3,j}$, $\omega'_{4,j}$ defined in Supplemental Material S8 (Box C) depending only on the haplotype and local ancestry matrices.

The joint distribution of local ancestry average causal effects allows us to compare average causal effects by local ancestry within an admixed population, similar to earlier studies. We follow the approach of Hou et al. (2023), which uses the correlation parameter of the statistical model to measure the similarity of causal effect. Denote the **L**ocal **A**ncestry **A**verage causal effect **Cor**relation (LAACor) for a variant $j$ under the global and the local models by LAACor$_j'^{\text{Glo}}$ and LAACor$_j'^{\text{Loc}}$ respectively. While it follows immediately from Eq. (2) that LAACor$_j'^{\text{Loc}} = \rho$, for the global model following Proposition 1 LAACor$_j'^{\text{Glo}} = w'_j/\sqrt{u'_j v'_j}$. Amalgamating correlations across all variants $j$, we can define genome-wide local ancestry average causal effect correlations ($\overline{\text{LAACor}'_{\text{Glo}}}$ and $\overline{\text{LAACor}'_{\text{Loc}}}$) and local ancestry average tagging effect correlations ($\overline{\text{LAACor}_{\text{Glo}}}$ and $\overline{\text{LAACor}_{\text{Loc}}}$), which are distributional parameters defined across all variant-specific local ancestry average causal effects. Supplemental Material Eqs. (S25) and (S26) describe analytical formulae for these quantities.

We compute $\overline{\text{LAACor}'_{\text{Glo}}}$, $\overline{\text{LAACor}'_{\text{Loc}}}$, $\overline{\text{LAACor}_{\text{Glo}}}$ and $\overline{\text{LAACor}_{\text{Loc}}}$ using real admixed genotype and local ancestry data, and calculate empirical correlations from simulated tagging and causal effects under Eqs. (2) and (11) (see Supplemental Material S8 for simulation details). We find that the global model implies high average causal effect correlations: $\overline{\text{LAACor}'_{\text{Glo}}}$ ranges from 0.989 to 1 as $\rho$ ranges from 0.2 to 1, with $\overline{\text{LAACor}'_{\text{Glo}}} \geq 0.998$ whenever $\rho \geq 0.9$. We plot pairs of points $(\rho, \overline{\text{LAACor}'_{\text{Glo}}})$ (Fig. 2), which illustrates that low causal effect correlation $\rho$ across continental ancestries can be consistent with a very high local ancestry average causal effect correlation. As reported in Supplemental Material Eq. (S25), the local model average causal effect correlation is on average just the causal effect correlation $\rho$. We observe this empirically in Fig. 2, where the points $(\rho, \overline{\text{LAACor}'_{\text{Loc}}})$ lie close to $y = x$. In particular, when $\rho$ itself is high ($\rho \geq 0.9$), both the local

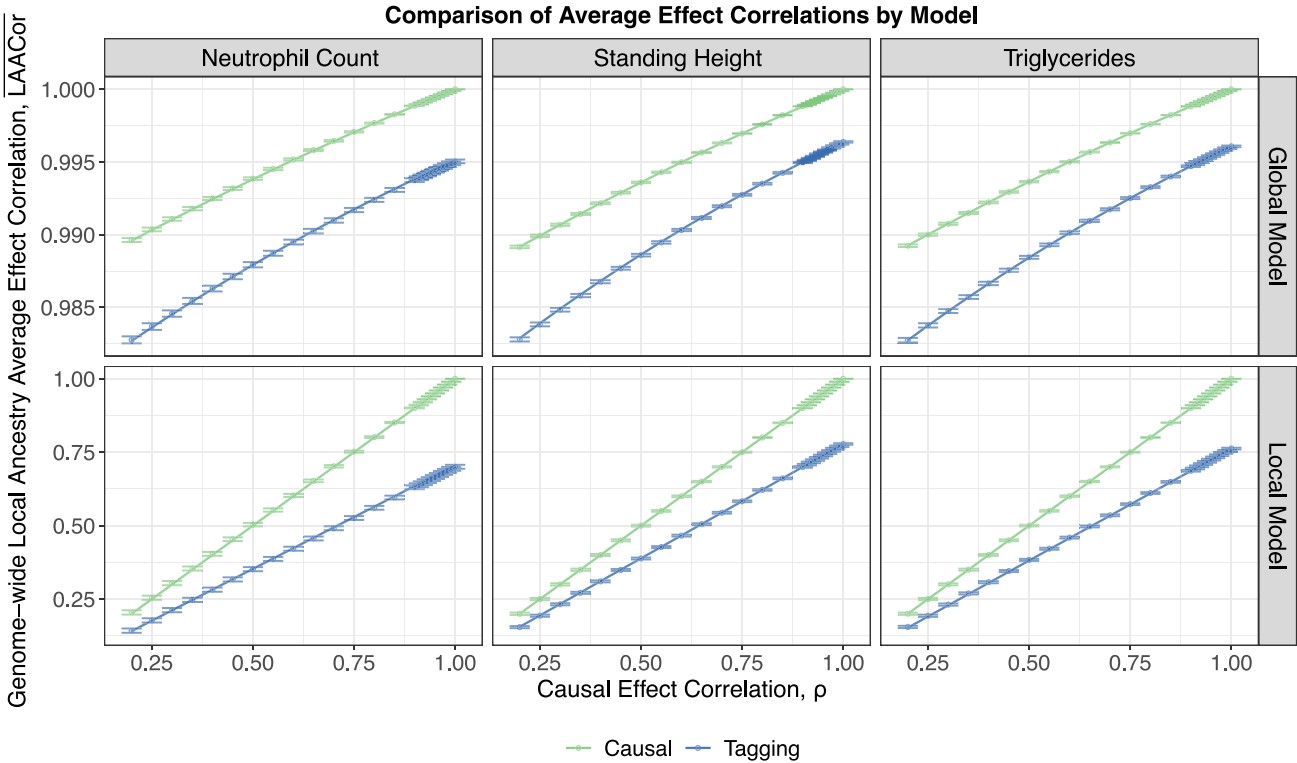

**Fig. 2.** Local ancestry average causal and tagging effect correlations predicted by the global and local models as a function of base model parameter $\rho$, using putatively causal and tagging variants for neutrophil count, height and triglycerides. Causal and tagging effect sizes are simulated under Eqs. (2) and (11), with variant-specific local ancestry average effects calculated as described in Individual and Average Effects. Genome-wide average effect correlations are obtained as empirical correlations between the vectors of variant-specific local ancestry average effects, and approximate 95% confidence intervals are shown. Curves correspond to the expected genome-wide local ancestry average causal effect correlations ($\overline{\text{LAACor}'}$ and $\overline{\text{LAACor}}$), distributional parameters described in Global model implies continuous distribution of individual level effect sizes, but still predicts highly similar average causal effects. The minima of tagging variant distributional parameters ($\overline{\text{LAACor}}_{\text{Glo}}$ and $\overline{\text{LAACor}}_{\text{Loc}}$) across values of $\rho \in [0.9, 1]$ are 0.994 ($\overline{\text{LAACor}}_{\text{Glo}}$) and 0.630 ($\overline{\text{LAACor}}_{\text{Loc}}$) for neutrophil count; 0.995 ($\overline{\text{LAACor}}_{\text{Glo}}$) and 0.700 ($\overline{\text{LAACor}}_{\text{Loc}}$) for height; and 0.995 ($\overline{\text{LAACor}}_{\text{Glo}}$) and 0.690 ($\overline{\text{LAACor}}_{\text{Loc}}$) for triglycerides.

and global models yield average causal effect correlations that are at least 0.9 (see Supplementary Figs. S19 and S20 for results across all six phenotypes).

Focusing next on average tagging effects, both models produce lower average effect correlations. As reported in Supplemental Material S8, the shrinkage in average effect correlation under the local model is driven by differences in LD between the two ancestries, quantified explicitly by the cosine similarity between the vectors of ancestry-specific LD (Supplemental Material Eq. (S27)). This is illustrated by the gap between the blue and green curves in the bottom row of Fig. 2. However, the global model still implies high average tagging effect correlations: $\overline{\text{LAACor}}_{\text{Glo}}$ ranges from 0.983 to 0.996 as $\rho$ ranges from 0.2 to 1, with $\overline{\text{LAACor}}_{\text{Glo}} \geq 0.994$ whenever $\rho \geq 0.9$ (top row of Fig. 2). Nevertheless, when $\rho \geq 0.9$ both models produce considerably high average tagging effect correlation: $\overline{\text{LAACor}}_{\text{Glo}} \geq 0.9$ and $\overline{\text{LAACor}}_{\text{Loc}} \geq 0.63$.

It may be surprising that weakly correlated causal effect sizes across ancestrally distinct populations can produce highly correlated local ancestry-averaged causal and tagging effect sizes, but this is the consequence of single-locus local ancestry being a poor predictor of global ancestry at the individual level. That is, the expected global ancestry conditioned on African local ancestry at a single locus is very close to the expected global ancestry conditioned on European local ancestry at that locus. Consequently, the average effect sizes conditioned on each local ancestry—which depend linearly on the average global ancestry conditioned

on local ancestry (see Eq. (7))—should also be very close (see dotted vertical lines in top density plot of Fig. 1d). In an extreme scenario where all individuals share a common global African ancestry, they are assigned the same effect size at each locus regardless of the local ancestry, and so the correlation conditioned on local ancestries is 1. Inter-individual variability in global ancestry lowers the correlation, but empirically this effect is small.

Using height as an example, Hou et al. (2023) estimated the average causal effect correlation parameter ($r_{\text{admix}}$, analogous to our $\overline{\text{LAACor}'}$) to be 0.94. This estimate is consistent with our global model, even for small values of $\rho$. However, the local model requires $\rho$ to also be 0.94 in order for such a high estimate of $r_{\text{admix}}$ to be observed empirically. Thus, our analysis of causal effects implies it is not possible to both have $\rho$ moderate to small, and also have the local model be a good fit to the genetic architecture of height.

## Partial polygenic scores distinguish models when causal variants and effects are known

If causal variants and their effect sizes are known, then polygenic scores (Eqs. (12) and (13)) are computed directly on the causal variants. We recall that all polygenic scores are constructed using effect sizes estimated from a population with only European ancestry. We show mathematically that, under simplifying assumptions of the distribution of local ancestries and allele frequencies, the behavior of the total polygenic score is

indistinguishable between the two models, but this is not the case for partial polygenic scores.

**Proposition 2**(Behavior of Causal Variant PGSs Under the Local and Global Models). Let the expected squared phenotype correlations with partial polygenic score and total polygenic score under the local model (resp., global model) be denoted by $\mathbb{E}_{\text{Loc}}[Cor^2(\text{ParPGS}, y)]$ and $\mathbb{E}_{\text{Loc}}[Cor^2(\text{TotPGS}, y)]$ (resp., $\mathbb{E}_{\text{Glo}}[Cor^2(\text{ParPGS}, y)]$ and $\mathbb{E}_{\text{Glo}}[Cor^2(\text{TotPGS}, y)]$). Let $\bar{a}$ denote the average African global ancestry in the admixed African American cohort, and $r^2$ and $\rho$ denote parameters controlling the variance explained by causal effects and the cross-ancestry correlation of genetic effects, as described in Eq. (2). Furthermore, assume that these conditions hold (formal treatment provided in Supplemental Material S5).

A) Ancestry-specific allele frequencies are identical across markers (but not necessarily between ancestries).
B) Distribution of local ancestries is the same across individuals.
C) Ancestry assignment is identical between two haplotypes of a diploid individual.
D) At each marker, the difference between ancestry-specific allele frequencies is small enough.
E) Distribution of alleles is approximately uncorrelated between two haplotypes at a diploid locus.

Then the following are true.

1) The squared phenotype correlations with total polygenic score are approximately equal between the two models.

$$\mathbb{E}_{\text{Glo}}[Cor^2(\text{TotPGS}, y)] \approx r^2(1 - \bar{a} + \rho\bar{a})^2 \approx \mathbb{E}_{\text{Loc}}[Cor^2(\text{TotPGS}, y)].$$

2) Under the global model the squared phenotype correlation with partial polygenic score is approximately a cubic function of global African ancestry.

$$\mathbb{E}_{\text{Glo}}[Cor^2(\text{ParPGS}, y)] \approx r^2(1 - \bar{a})(1 - \bar{a} + \rho\bar{a})^2.$$

3) Under the local model the squared phenotype correlation with partial polygenic score is approximately a linear function of global African ancestry.

$$\mathbb{E}_{\text{Loc}}[Cor^2(\text{ParPGS}, y)] \approx r^2(1 - \bar{a}).$$

To prove Proposition 2, we derive formulas for expected squared correlations for our model without any simplifying assumptions (Supplemental Material S4; see also Supplementary Tables S3–S5), and subsequently applied conditions (A)–(E) to arrive at the expressions printed in 1, 2 and 3 (Supplemental Material S6). Proposition 2 allows us to examine the behavior of ParPGS and TotPGS as a function of the model parameters $r^2$, $\rho$ and $\bar{a}$, under either model. As Fig. 3a shows, TotPGS scales quadratically in the average African global ancestry of the cohort, with larger values of $\rho$ forcing the relationship to become increasingly linear. This is not surprising, given that as $\rho$ approaches 1, $1 - \rho$ approaches 0 and so the quantity $r^2(1 - \bar{a} + \rho\bar{a})^2 = r^2[1 - (1 - \rho)\bar{a}]^2 \approx r^2[1 - 2(1 - \rho)\bar{a}]$—which is linear in $\bar{a}$. What may be more surprising is that this relationship is shared between the local and global models. To see why, we may think of the PGS-phenotype correlation in an admixed person as a mixture: a European piece with full strength ($r$) plus an African piece that is attenuated by how transferable the effects are ($\rho$). By weighing those pieces by how much of the genome sits in each ancestry—on average, $(1 - \bar{a})$ and $\bar{a}$ respectively, regardless of the model assumed—we obtain the total strength (that is, the PGS-phenotype correlation) $r(1 - \bar{a}) + r\rho\bar{a} = r(1 - \bar{a} + \rho\bar{a})$, the square of which gives the expression for predictive power.

Focusing next on ParPGS, we see that under the global model, it cannot be more predictive than TotPGS—as illustrated in Fig. 3a, by the black curve always lying above the orange curve for paired orange and black curves. Proposition 2 quantifies this "prediction shrinkage factor" as the global European ancestry, $1 - \bar{a}$ (in particular, the gap does not depend on either $r^2$ or $\rho$). Under the local model, however, the prediction shrinkage factor has an inverse relationship with causal effect correlation $\rho$: holding the quantities $\bar{a}$ and $r^2$ fixed, as $\rho$ increases from 0 to 1, the predictive power of TotPGS increases from $r^2(1 - \bar{a})^2$ to $r^2$, and so the predictive power of ParPGS relative to this quantity decreases. This is a consequence of the ratio $\mathbb{E}_{\text{Loc}}[Cor^2(\text{ParPGS}, y)]/\mathbb{E}_{\text{Loc}}[Cor^2(\text{TotPGS}, y)] \approx (1 - \bar{a})/(1 - \bar{a} + \rho\bar{a})^2$ being a decreasing function of $\rho$: defining $f(\rho, \bar{a}) = (1 - \bar{a})/(1 - \bar{a} + \rho\bar{a})^2$, we have $\partial f/\partial\rho = -2\bar{a}(1 - \bar{a})/(1 - \bar{a} + \rho\bar{a})^3$, which is less than zero for any $\bar{a} \in (0, 1)$ and $\rho \in [0, 1]$. This analytical observation also shows that ParPGS can be more predictive than TotPGS (cyan curve lying above black curve in Fig. 3a). This occurs when the causal effect correlation between distinct ancestrally homogeneous populations, $\rho$, is less than 0.5, and the global European ancestry is large enough (more precisely, when $1 - \bar{a} \geq [\rho/(1 - \rho)]^2$). The intuition for this is that, if ancestry-specific causal effects are sufficiently different (driven by *cis* interactions, for example) and the admixed cohort inherited largely European haplotypes, then weighing the very few causal variants carrying African local ancestries with European ancestry-specific effects would hinder rather than boost prediction.

Crucially, and from a more practical viewpoint, we verify that our analytical results in Proposition 2 hold qualitatively even without making any simplifying assumptions. Simulating phenotypes from 50 different seeds of approximately independent phased genotype markers in admixed PMBB individuals, we varied both $r^2$ and $\rho$ to capture a variety of genetic architectures. We confirm that TotPGS does not differentiate between the two models while ParPGS does, with small standard errors observed across seed-specific average squared correlation (Fig. 3b; see Supplementary Figs. S2 and S3 for results specific to each choice of $(r^2, \rho)$). Finally, Proposition 2 also reveals a key limitation to the use of ParPGS to distinguish the local and global models: when causal effect correlation between ancestrally homogeneous populations is high ($\rho \approx 1$), local and global models are difficult to distinguish, since both behave roughly linearly in global African ancestry. This makes sense, since if causal effects are highly correlated, then effect sizes under the global and local models are also highly similar, and so polygenic scores computed on causal variants would be indistinguishable.

## Use of tagging variants limits distinguishability of local and global models

In general, we only have access to tagging variants ascertained from GWAS summary statistics. Similar to Proposition 2, we derive approximations to the behavior of total and partial polygenic scores under the two models, and summarize them under Proposition S1 in Supplemental Material S6. Unlike the case where causal variants are known, the approximations are not reducible to "summary" quantities, and instead depend on LD between tagging and causal variants, cross-ancestry

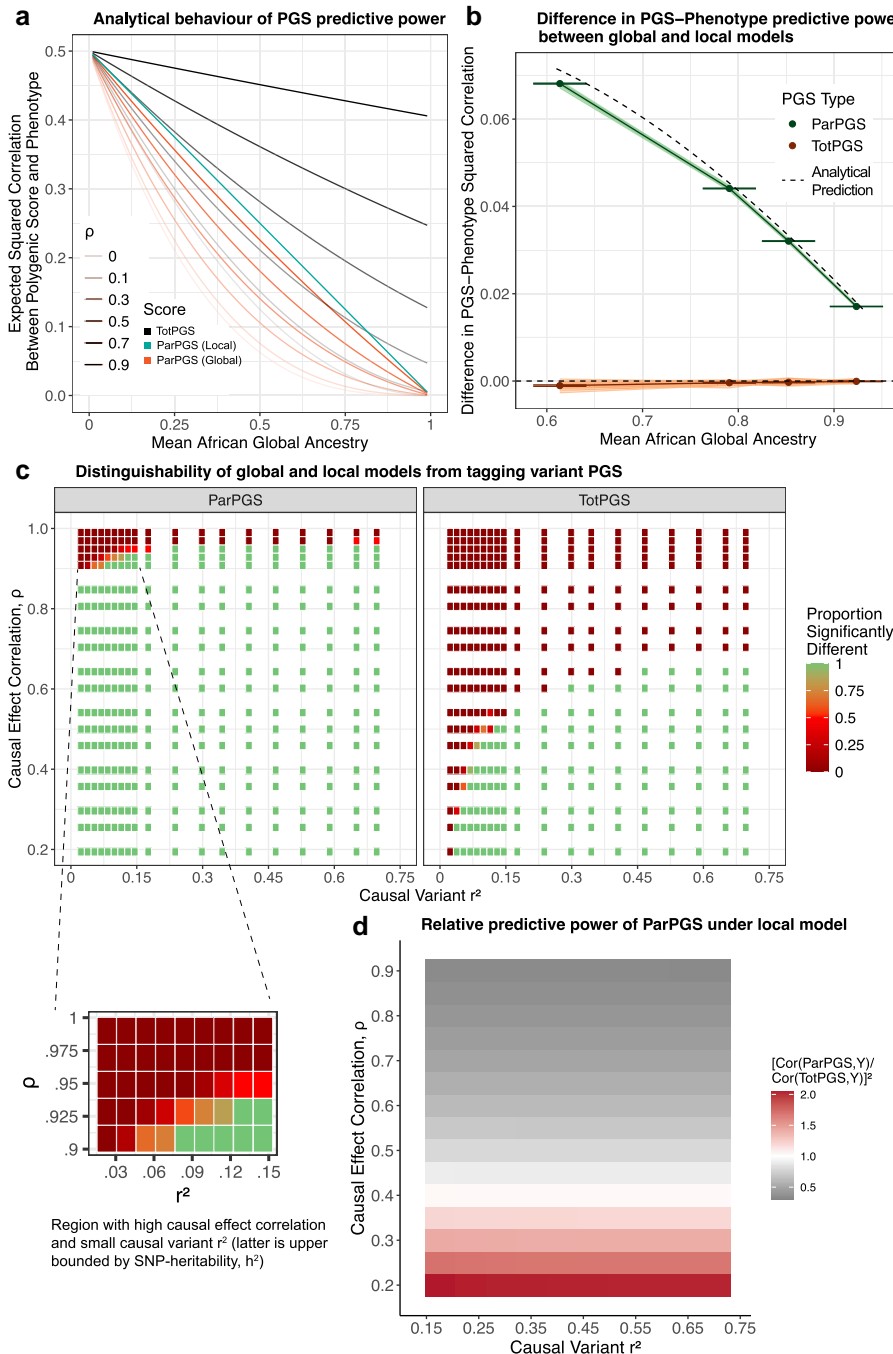

**Fig. 3.** a) Relationship between performance of standard (TotPGS) and partial polygenic scores (ParPGS), measured by squared correlation with phenotype, as predicted by theory in the scenario where causal variants are known. Curves are plotted based on Proposition 2. b) Comparison of ParPGS and TotPGS correlations with phenotype, between the local and global models. For each seed and configuration of simulation parameters, differences in PGS-phenotype correlations between the two models were computed. The dark points and lines show differences averaged across all seeds and simulation configurations, while the light lines show seed-specific differences averaged across all seed-specific simulation configurations. The approximate 95% confidence intervals for the dark points were calculated from standard errors across seed-specific differences (range of standard errors across all quantiles: $[3.8 \times 10^{-5}, 8.0 \times 10^{-5}]$). Dashed lines are analytical predictions based on Proposition 2. c) Distinguishability of global and local models from tagging variant PGS, based on performing Wilcoxon rank sum tests on simulated phenotypes under each model for six phenotype-specific sets of causal and tagging variants. We use $\alpha = 0.01$ as a cutoff for significance, and compute the proportion of comparisons deemed significant within each $(\rho, r^2)$ bin. Figure inset zooms into the top left parameter range $0.9 \leq \rho \leq 1$ and $0.02 \leq r^2 \leq 0.15$. d) Predictive power of tagging variant ParPGS relative to tagging variant TotPGS, for phenotypes simulated using height-specific causal and tagging variants.

correlation of causal variants, and allele frequencies of causal variants.

While the quantities described above are not typically available in practice, we nevertheless performed simulations to better understand the behavior of total and partial polygenic scores and to evaluate the approximations derived in Proposition S1. Specifically, using putatively assigned causal and tagging variants based on phenotype-specific summary statistics

(Simulation Study 2), we simulated synthetic versions of six phenotypes while varying both $r^2$ and $\rho$—in particular choosing $r^2$ to reflect realistic ranges for each phenotype.

As one would expect, the use of tagging variants reduces the predictive power of both ParPGS and TotPGS (Supplementary Figs. S4–S15). Unlike the scenario where causal variants are known, both ParPGS and TotPGS can differentiate the two models, with the approximations described in Proposition S1 holding reasonably well for each phenotype (Supplementary Figs. S4–S15). However, the ability of each polygenic score to distinguish the models also depends on the parameters $r^2$ and $\rho$, with $\rho$ more strongly constraining the distinguishability (Fig. 3c). Tagging variant TotPGS can differentiate the models whenever $\rho$ is small and $r^2$ is not too small because, provided that European effect sizes are predictive in European ancestry backgrounds (i.e. $r^2$ not too small), the LD heterogeneity between ancestries and across markers attenuates the predictive power of European effect sizes differently between the two models, which is sufficient to dominate the attenuation by cross-ancestry transferability of effect sizes should the latter not be too large (i.e. $\rho$ is small). Similar to the causal variants known scenario, ParPGS generally has a stronger ability to distinguish the local and global models, unless causal effect correlations are very high ($\rho \geq 0.9$) and the signal-to-noise ratio is also low ($r^2 \leq 0.15$); see Fig. 3c inset. Furthermore, the local model permits ParPGS having more predictive power than TotPGS, although the precise range of values of $\rho$ and $\overline{a}$ for which this occurs is less clear. Using simulations with height causal and tagging variants as a case study, we find empirically that $\overline{a} \approx 0.8$, with $\rho \in [0, 0.4]$ being the approximate range of values for which the relative predictive power of ParPGS exceeds 1; see Fig. 3d. (For comparison: solving $1 - \overline{a} \geq [\rho/(1 - \rho)]^2$ in the causal variants known scenario yields $\rho \leq \sqrt{0.2}/(1 + \sqrt{0.2})$, or $\rho \in [0, 0.31]$.) Similar to the causal variants known scenario, the global model always predicts ParPGS having no more predictive power than TotPGS, although the prediction shrinkage factor generally lies above $1 - \overline{a}$ (Supplementary Fig. S16). Another difference is that the relative predictive power of ParPGS under the global model now decreases slightly as $\rho$ increases (Supplementary Fig. S16)—unlike the causal variants known scenario where $\rho$ does not affect this quantity (Partial polygenic scores distinguish models when causal variants and effects are known). Under the local model, however, just like the causal variants known scenario we observe an inverse relationship between causal effect correlation $\rho$ and relative predictive power of ParPGS (Supplementary Fig. S17).

To clarify how the use of tagging variants hinders model distinguishability, we use height again as an example. In simulations of height-specific putatively causal and tagging variants with $\rho = 0.5$ (Supplementary Fig. S10), the local model ParPGS predictions are almost indistinguishable from the global model ParPGS prediction if causal variants were known (e.g. both predict a squared correlation of 0.03 when $r^2 = 0.4$). Thus, without accounting for causal-tagging LD and allele frequency differences between ancestries, we may mistakenly find the global model to be a good fit, even though in reality the local model can explain the observed behavior of the polygenic score. Another challenge hindering model distinguishability is that, if we expect the genetic effects of complex traits to interpolate between the local and global models, then we have to estimate the contribution of each model to the observed summary statistics, including the relative predictive power of the partial polygenic score. As reported above, an observed relative predictive power >1 would imply some contribution by the local model, but we cannot immediately rule out global model contributions entirely (similarly, relative predictive power ≤1 may also be consistent with entirely local model contributions). Estimating the contributions of each model would require estimating parameters of our base model ($\rho$ and $r^2$), which is hindered by the lack of causal-tagging variant LD information.

## Discussion

We have introduced models for how individual variant effects in admixed individuals may be influenced by ancestry. Other studies have evaluated differences in variant effects across ancestral contexts (Fig. 1a and Supplementary Table S1). According to Hou et al. (2023) and Hu et al. (2025), average causal variant effects are highly similar between ancestries (for example, a correlation of 0.94 in genetic effects for height between European and African ancestry backgrounds). The simplest explanation is that individual causal effects are in fact identical between ancestries. However, we have shown that this observation is also compatible with variable individual causal effects that vary continuously with global ancestry. This is equivalent to the possibility, in stratified randomized experiments, of heterogeneity in individual treatment effect despite homogeneity in average treatment effect between strata (Hernán and Robins 2020, Chapter 1). On the other hand, we find that a model in which individual causal effects vary with local ancestry is inconsistent with the observation of highly correlated average causal effects.

We show that when causal variants are known, the difference between the predictive power of total and partial polygenic scores can be used to directly test whether variant effects depend on local or on global ancestry. However, in practice, when causal variants are unknown, the use of tagging variants reduces the predictive power of both standard and partial polygenic scores, with differences in LD between causal and tagging variants across ancestries confounding analyses of polygenic score behavior under the two models (so that the models cannot practically be distinguished). In principle, a subset of reliably fine-mapped causal variants could be used to directly test the two models.

Our work contributes to ongoing discussions about the genetic architecture of complex traits and their implications for polygenic prediction. Almost all complex traits are highly polygenic, but polygenic scores trained on one ancestrally homogeneous population do not port well to other ancestries. Some authors have proposed that most genetic effects are determined by numerous peripheral genes (the omnigenic model; Boyle et al. 2017; Liu et al. 2019; Gazal et al. 2022). Given that direct effects on peripheral genes depend on environmental exposures and genetic drift, which differ between ancestries, one might expect genetic effects to exhibit interactions with global ancestry (Mathieson 2021), corresponding to our global model. In this case, effect sizes inferred from one cohort would port poorly to cohorts that differ in ancestry and environmental exposures, even after accounting for differences in linkage disequilibrium (LD) and allele frequencies. Conversely, other work suggests that ancestry representation is the core issue. In this view, causal variants share similar effects across ancestrally divergent cohorts, but overrepresentation of European samples has resulted in bias of genome-wide association studies toward European cohorts (Pasaniuc et al. 2011; Hou et al. 2023; Hu et al. 2025). Polygenic scores built from variants ascertained in these cohorts would predict poorly in another ancestrally divergent population, largely due to differences in LD patterns and allele frequencies of causal variants. Our work clarifies the intricate relationships between the fundamental assumptions in statistical models of effect sizes, individual-level genetic effects, and polygenic risk prediction, implying that

gene-by-ancestry interactions should not be rejected even if average causal effects are highly similar across ancestries and ancestry representation in Biobanks is lacking. Instead, the impacts of LD and of causal effect correlation across ancestries on estimable genetic effects must be disentangled, to determine the relative influence of local and global ancestries on polygenic score transferability into admixed groups.

Equally important, gene-by-ancestry interactions do not imply that causal effects differ discretely between ancestries. On the contrary, such statistical interactions arise from inter-individual variability in causal effects, itself a consequence of the huge diversity in patterns of genetic variation and longitudinal environmental exposures observed between any pair of individuals. Gene-by-ancestry interactions could reflect extensive gene-by-gene (G×G) interactions, but could also result from gene-by-environment interactions (G×E) if environment is correlated with ancestry among admixed individuals (Park et al. 2018; Patel et al. 2022). Our models do not directly distinguish between these possibilities. If population average effects really are identical between European and African Americans, then gene–gene interactions are a more plausible explanation, because it is unlikely for an environmental variable to vary with ancestry among African Americans but still have the same mean value in European and African Americans. On the other hand, if population average effects differ then G×G and G×E are both possible explanations. In any case, our work supports ongoing efforts to better understand interactions between genes and ancestry or environmental exposures (e.g. Westerman and Sofer 2024).

To conclude, our work demonstrates how gene-by-ancestry interactions and individual-level effect heterogeneity may be hidden in standard polygenic score representations of complex traits. We advocate fine-mapping and other approaches that will reliably measure linkage between tagging and causal variants as well as interactions of causal variant effects with ancestry. These steps will support strategies that improve the prediction of individual-level polygenic risk to make precision medicine inclusive to all.

## Data availability

Code to run simulations is available at https://github.com/mathilab/GxA_Interaction. Individual-level PMBB data are not publicly available, but the code repository includes summary-level data sufficient to recreate the results and scripts for reproducing the Main Text figures. Supplemental material (PDF file) available online.

Supplemental material available at GENETICS online.

## Acknowledgments

We gratefully acknowledge PMBB participants for their contributions, without whom this research would not have been possible. We thank Regeneron Genetics Center (RGC) for PMBB data generation, Andy Dahl for technical comments, Arslan Zaidi for help with analyzing PMBB data and the members of the labs of Iain Mathieson and Ziyue Gao for helpful discussions.

## Funding

This research was supported by National Institute of General Medical Sciences (NIGMS) award number R35GM133708 and National Human Genome Research Institute (NHGRI) award number 5T32HG009495. The content is solely the responsibility of the authors and does not necessarily represent the official views of the National Institutes of Health.

## Conflict of interests

None.

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

*Editor: H. Mostafavi*