## [Peer Review File · Genetics]

Hidden structure in polygenic scores and the challenge of disentangling ancestry interactions in admixed populations

Alan Aw, Ravi Mandla, Zhuozheng Shi, Bogdan Pasaniuc, and Iain Mathieson

NOTE: The reviews and decision letters are unedited and appear as submitted by the reviewers.

In extremely rare instances and as determined by a Senior Editor or the EIC, portions of a review may be redacted. If a review is signed, the reviewer has agreed to no longer remain anonymous.

The review history appears in chronological order.

Review Timeline:

Submission Date:	2025-06-27
Editorial Decision:	2025-08-15
Resubmission Received:	2025-09-01
Accepted:	2025-09-18

August 15, 2025

GENETICS-2025-308316

Hidden structure in polygenic scores and the challenge of disentangling ancestry interactions in admixed populations

Dear Dr. Aw:

Two experts in the field have reviewed your manuscript, and I have read it as well. I am pleased to inform you that, with minor revisions, it is potentially suitable for publication in GENETICS. The reviewers have comments and concerns that need to be addressed in a revised manuscript. You can read their reviews at the end of this email.

In brief, both reviewers expressed enthusiasm for the paper's key results, which help reconcile the apparent contradiction between the similarity of genetic effects across ancestries even with the possibility of pervasive interactions. They have also offered several constructive suggestions, all of which we believe can be addressed.

We particularly highlight the shared request from both reviewers to add more verbal intuition for key results and assumptions. Specifically: (i) as noted by Reviewer #1, please provide additional intuition alongside the mathematical presentation on why and how mean effects conditional on global ancestry can be similar even when the true causal variants differ substantially, and on the performance of polygenic scores based on European effect sizes; and (ii) given Reviewer #2's positive response to the supplementary note on this topic, we encourage you to incorporate some of the rationale for demeaning genotypes into the main text so that these insights are more visible to readers.

In addition, please address the request for a fuller discussion of the biological interpretation and implications of the "global" model, as well as clarification and justification of the simulation methods.

We look forward to receiving your revised manuscript. Please let the editorial office know approximately how long you expect to need for revisions.

Upon resubmission, please include:

1. A clean version of your manuscript;
2. A marked version of your manuscript in which you highlight significant revisions carried out in response to the major points raised by the editor/reviewers (track changes is acceptable if preferred);
3. A detailed response to the editor's/reviewers' comments and to the concerns listed above. Please reference line numbers in this response to aid the editors.

Additionally, please ensure that your resubmission is formatted for GENETICS.

<https://academic.oup.com/genetics/pages/general-instructions>

Follow this link to submit the revised manuscript: Link Not Available

Sincerely,

Hakhamanesh Mostafavi
Associate Editor
GENETICS

Approved by:
Anthony Long
Senior Editor
GENETICS

Reviewer #1 :

This paper considers different ways in which variant effect sizes can differ across ancestry groups, the downstream effects that these sources of effect-size heterogeneity have on effect size estimates in admixed populations, and methods that can be used in admixed populations to empirically distinguish these sources of effect-size heterogeneity.

The authors define two models by which ancestry can alter the effect of a variant: a "local" model where it is a variant's local

ancestry that determines its effect (via cis modifiers, for example), and a "global" model where the variant's effect in a given individual depends on that individual's genome-wide ancestry (which might pertain if there are many trans modifiers of the variant's effect, or under certain kinds of environmental inheritance and interaction).

They show mathematically and in simulated and real datasets that, under the global model, in an admixed sample, the average effects estimated for variants, conditional on local ancestry, can be highly correlated between the two possible local ancestries. This is true even when effects within the two ancestry groups are only weakly correlated, suggesting strong caution in interpreting the large correlations that have been found between local-ancestry-controlled effect-size estimates.

The authors also show that, when PGSs are constructed using effect sizes estimated in one ancestry group, trait prediction in an admixed population is approximately equally successful under the local and global models if the PGS is built across all loci (regardless of their local ancestry state). However, a PGS built only for those loci in an individual that have European local ancestry state predicts trait values better under the local model than under the global model, especially when effect sizes are strongly correlated between the two ancestry groups.

Overall, I found the results interesting and highly relevant to ongoing efforts to interpret similarities and differences between effect sizes estimated for different ancestry groups. The manuscript is well written, and I particularly appreciated the clarity of the Supplementary Information. I have only a few minor comments for improvement of the manuscript.

MINOR COMMENTS

-- One of the authors' main results is that, under their global model, there is a very strong correlation between average effects estimated conditional on European and African local ancestry. Indeed, even when the effects of variants in the source populations are only weakly correlated ($\rho = 0.2$), the effects of variants given African local ancestry and the effects given European local ancestry are nonetheless strongly correlated (averaging 0.989 for causal effects and 0.983 for tagging effects; page 11). This is a facially surprising result, and I think it would be useful for the authors to try to give the reader a better sense of its intuition. For example, is the intuition that local ancestry at a single locus is a poor predictor of global ancestry, and so the average global ancestry conditional on African local ancestry at locus j is very close to the average global ancestry conditional on European local ancestry at locus j ? Since the average causal effect conditional on local ancestry is a linear function of the average global ancestry conditional on the local ancestry (Eq. 7), it would seem that this would lead the average causal effects conditional on the two possible local ancestries to be basically the same.

-- Similarly, the authors find that the performance of a European effect-size-based polygenic score calculated over the entire genotype (regardless of local ancestry) performs approximately equally, in terms of trait prediction, whether causal effect sizes follow the local or global model. What is the intuition for this result?

-- page 5: In the authors' "global" model (Eq. 7), a specific variant's effect in a given individual depends on that individual's genome-wide ancestry proportions. This is an appealingly simple model which, as the authors say, summarizes the contribution of a great many trans-acting modifiers of the variant's effect, or perhaps GxE. However, I think it might be worthwhile for the authors to dedicate a little more discussion---either here or in the Discussion section itself---of the mechanisms that could underlie such a model. For example, if the effect of global ancestry on the variant's effect is via trans-acting modifiers, should we think of these modifiers as fixed differences between the ancestries? Or are they segregating within each ancestry group, but at different frequencies? If the latter, then, since these modifiers have a causal effect on the trait, should we think of them as among the set of p causal variants? If so, why do we not need to explicitly model the effect of a variant in a given individual as a function of their genotype at all causal loci? If, on the other hand, the dependence of a variant's effect on global ancestry is via GxE, this requires the environment (and its interaction with the variant) to be linear in the individual's ancestry, which would be a very specific kind of environmental inheritance and GxE, meriting some discussion.

-- page 7: Eqs. (9) and (10) assume that effect sizes are estimated without bias, and therefore that there is no confounding, which should be noted.

-- SI page 32: Under what conditions are the approximations in the derivations of LAACor'Loc and LAACor'Glo valid?

-- page 11: For the conditions under which Proposition 4.2 holds, we are directed to Supplementary Section S5 (page 20 of the SI, bullets (A)-(E)). However, some of these conditions are very stringent (for example, equal within-ancestry allele frequencies across markers, identical ancestry assignment across haplotypes in a diploid individual), and therefore should be included explicitly in the statement of the Proposition.

-- page 12: In the caption of Figure 2, it says that "approximate 95% confidence intervals are shown" for the genome-wide average effect correlations displayed, but I cannot see these confidence intervals in the figure.

TYPOS/WORDING

page 4: "whose haplotypes are columns j and $j+p$ " -- In what sense are these haplotypes? The columns give the two haploid genotypes at causal locus j in the sampled individuals.

SI page 2: diploid  diploid

Reviewer #2 :

Overall, I think this paper makes an important theoretical contribution to an area of research that has become especially salient for understanding and improving polygenic score portability. It provides a biologically realistic framework for thinking about causal effect differences across populations, which is something that has been missing from the literature thus far.

Below I outline some specific questions, along with some suggestions for clarifying the models and assumptions in the main text.

- If phenotypes and effect sizes are both being simulated, I'm unclear on why the GWAS summary statistics are needed for simulation study 2. Can causal and tag variant pairs not simply be simulated by sampling from all pairs of tightly linked variants? What benefit does it provide to condition on GWAS hits?
- Similarly, I'm curious about the choice to approximately estimate r^2 and the number of causal variants through a proxy analysis of polygenic score performance. Is there a reason why published heritability or polygenicity estimates wouldn't be suitable for these simulations? By comparison, the approach used in this paper seems more challenging to perform and harder to justify.
- A few questions on notation in Supplement S2: is it correct to say that b is the true individual-level effect and β is the group-averaged marginal effect? Also, are the β s in this section corresponding to causal β s? (They are missing the apostrophe used to denote causal variant effects elsewhere in the manuscript.)
- Under the global model, does LAACor depend on the distribution of global ancestry? It's hard to tell from the expressions in Supplement S8, but intuitively, it seems that it could, and it would be interesting to know if this is the case.
- It might be nice to give some more intuition for the comparison of causal variant PGS and tag variant PGS - it was difficult to think about why causal variant TotPGS is identical for local and global models, but tag variant TotPGS can distinguish between them.
- It seems that there are parameter regimes where both ParPGS and TotPGS can distinguish local and global models, and so from Figures 3C and S4-15, it was not obvious to me why the ratio of tagging variant ParPGS and TotPGS could not distinguish local and global models. A tagging variant version of 3A might be nice to include in the supplement, or else the authors could provide some more intuition for the reader.
- In Figure 3B, is there no uncertainty in the correlation estimated from the different seeds? I was confused by the orientation of the standard error bars.
- I found it somewhat difficult to interpret the various β s on a first pass through the manuscript. In section 2.1, it could be helpful to define the ancestry-specific β s before writing out the distribution. Otherwise, I think it is only stated explicitly in Supplement S2 that the ancestry-specific β s are the average causal effect in a population that is "homogeneous" for that particular ancestry. It may potentially be helpful to also include some of the explicit generative models from Supplement S2 in the main text to make the global and local models more transparent to readers.
- The paper uses the phrase "gene-ancestry interaction" in a few places, and I'm not always sure what that means. I assume interaction is being used in the statistical sense, but it's a bit confusing because the paper also talks about interactions in the biological sense. I also think the nomenclature "gene-ancestry interaction" verges somewhat closer to implying causal effects differ discretely between ancestries, which one wants to avoid. (It's especially notable because in other aspects, the paper does a very good job of emphasizing individual-level variation in causal effects and de-emphasizing continental ancestries as discrete groups.)
- Figure 1D was somewhat confusing. While I think I understand what the authors are trying to convey, I don't know what specifically is being plotted in the global model diagram and the associated inset.
- If I understood correctly, LAACor is defined as the correlation within an admixed population, and it would be good to make that explicit in 4.1. In the main text, I think this is only stated implicitly via reference to Hou et al.
- The note on demeaning genotypes in Supplement S1 is quite nice, I hadn't thought of that.

- It would be helpful to reiterate at the start of 4.2 that under this framework, PGS are constructed using betas estimated from a population with only European ancestry (this is explained earlier, in 2.3, but easy to lose track of by the time one gets to 4.2).

- I frankly thought the authors were selling themselves short with the sentence in the Discussion "One limitation of our study is that our approach is indirect, relying on a quantitative comparison to other work". Theoretical work can so often help us understand empirical results, and that is true of this paper as well.

Roshni Patel

Response to Reviews

Associate Editor

We particularly highlight the shared request from both reviewers to add more verbal intuition for key results and assumptions. Specifically: (i) as noted by Reviewer #1, please provide additional intuition alongside the mathematical presentation on why and how mean effects conditional on global ancestry can be similar even when the true causal variants differ substantially, and on the performance of polygenic scores based on European effect sizes; and (ii) given Reviewer #2's positive response to the supplementary note on this topic, we encourage you to incorporate some of the rationale for demeaning genotypes into the main text so that these insights are more visible to readers.

In addition, please address the request for a fuller discussion of the biological interpretation and implications of the "global" model, as well as clarification and justification of the simulation methods.

Upon resubmission, please include:

1. A clean version of your manuscript;
2. A marked version of your manuscript in which you highlight significant revisions carried out in response to the major points raised by the editor/reviewers (track changes is acceptable if preferred);
3. A detailed response to the editor's/reviewers' comments and to the concerns listed above. Please reference line numbers in this response to aid the editors.

We have addressed both (i) and (ii), as well as the request for a fuller discussion, per the Reviewers' comments.

On top of the revised clean manuscript, we have included a "marked" version showing tracked changes. Line numbers with respect to the tracked changes file are provided for easy reference. Especially for (ii), we have taken up your encouraging suggestion and included a brief elaboration in **Lines 95-99**.

We have also updated our Github repo (https://github.com/mathilab/GxA_Interaction) to reflect changes to our Main Text figures, thus maintaining full reproducibility of our results.

Reviewer 1

MINOR COMMENTS

1) One of the authors' main results is that, under their global model, there is a very strong correlation between average effects estimated conditional on European and African local ancestry. Indeed, even when the effects of variants in the source populations are only weakly correlated ($\rho = 0.2$), the effects of variants given African local ancestry and the effects given European local ancestry are nonetheless strongly correlated (averaging 0.989 for causal effects and 0.983 for tagging effects; page 11). This is a facially surprising result, and I think it would be useful for the authors to try to give the reader a better sense of its intuition. For example, is the intuition that local ancestry at a single locus is a poor predictor of global ancestry, and so the average global ancestry conditional on African local ancestry at locus j is very close to the average global ancestry conditional on European local ancestry at locus j ? Since the average causal effect conditional on local ancestry is a linear function of the average global ancestry conditional on the local ancestry (Eq. 7), it would seem that this would lead the average causal effects conditional on the two possible local ancestries to be basically the same.

Thank you very much for this suggestion, and for providing a clear explanation that we have incorporated into our Main Text (**Lines 341-352**). We also find the following intuition helpful: suppose in the extreme case that everyone has the same global ancestry (i.e., single-locus local ancestry is entirely uninformative about global ancestry). Then the global model assigns the same effect size to every individual at the locus, regardless of their local ancestry. Consequently, the correlation conditioned on local ancestries is 1. Inter-individual variability in global ancestry lowers the correlation, but empirically we do not observe it lowering the correlation considerably.

2) Similarly, the authors find that the performance of a European effect-size-based polygenic score calculated over the entire genotype (regardless of local ancestry) performs approximately equally, in terms of trait prediction, whether causal effect sizes follow the local or global model. What is the intuition for this result?

The intuition comes from observing that either model should lead to a PGS-phenotype correlation that is approximately the sum of two components: $r^*(1 - \bar{a}) + r^*\rho^*\bar{a}$. (Note: predictive power is measured by *squared* correlation, so the expression here is consistent with Proposition 4.2 Statement 1.) The first component, $r^*(1 - \bar{a})$, captures the predictive power in the European segments, whereas the second component, $r^*\rho^*\bar{a}$, captures the predictive power in the African segments, where ρ is an attenuation factor

depending on how transferable the causal effects are across ancestries. We can intuit this by considering a special case where all individuals share the same global ancestry, \bar{a} . Under the global model, effect sizes for an individual are a weighted combination of the European and African effect sizes, where \bar{a} is the weight, so using European effect sizes leads to a phenotype correlation that is similarly weighted, with an extra shrinkage factor ρ to account for the imperfect similarity between European and African effect sizes (as described by Eqs. 2-5 of the base model). Under the local model, effect sizes for an individual are either European or African, with the former occurring at a $(1 - \bar{a})$ fraction of the entire set of causal variants. Therefore, regardless of the model considered, the phenotype-genotype correlation is either r for a European-coloured locus (occurs at fraction $(1 - \bar{a})$) or $r*\rho$ for an African-coloured locus (occurs at fraction \bar{a}).

We have incorporated this reasoning into the Main Text (**Lines 394-400**).

3) page 5: In the authors' "global" model (Eq. 7), a specific variant's effect in a given individual depends on that individual's genome-wide ancestry proportions. This is an appealingly simple model which, as the authors say, summarizes the contribution of a great many trans-acting modifiers of the variant's effect, or perhaps GxE. However, I think it might be worthwhile for the authors to dedicate a little more discussion---either here or in the Discussion section itself---of the mechanisms that could underlie such a model. For example, if the effect of global ancestry on the variant's effect is via trans-acting modifiers, should we think of these modifiers as fixed differences between the ancestries? Or are they segregating within each ancestry group, but at different frequencies? If the latter, then, since these modifiers have a causal effect on the trait, should we think of them as among the set of p causal variants? If so, why do we not need to explicitly model the effect of a variant in a given individual as a function of their genotype at all causal loci? If, on the other hand, the dependence of a variant's effect on global ancestry is via GxE, this requires the environment (and its interaction with the variant) to be linear in the individual's ancestry, which would be a very specific kind of environmental inheritance and GxE, meriting some discussion.

Thank you for raising this important and valid point. Re: *trans*-acting modifiers, assuming that the biological mechanisms (comprising signaling pathways, circuits of expression control, etc.) are shared across all individuals, it would make more sense for the latter viewpoint — segregation but at different frequencies between ancestries — to underlie the global model. Without question these modifiers lead to causal interactions (we refer to Chapter 9 of Vanderweele's *Explanation in Causal Inference: methods for mediation and interaction* for terminology), though their relatively tinier effects and the computational burden they impose on genetic analyses are reasonable considerations

for their exclusion from the set of p variants in practice. Indeed, a natural implication is to somehow incorporate the set of all causal variants into statistical models of individual variant effect, and we see this as a potential direction for future projects that carry a predictive modeling flavour. An example of work converging onto this viewpoint is Choi et al., 2023 *PLoS Genetics* [PMID: 36749789]; though we think challenges abound in this direction. (For example, ensuring accurate ascertainment of pathways and efficiency of procedures for conditioning on relevant background genetic variation within the pathway to obtain unbiased variant effects.)

However, we showed in Supplement S2 that global ancestry is an approximate measure of effect modification by *trans*-modifiers. Our demonstration starts from a quantitative-genetic model of single-locus effect size, alongside many *trans*-modifier variants introducing causal interactions, which produces three distinct expected genetic effects — one for an individual carrying full European ancestry (β^{Eur}), one for an individual carrying full African ancestry (β^{Afr}), and one for a two-way African and European admixed individual (β^{Glo}). Expectations are taken over the distribution of alleles at the epistatic and focal loci, leading to terms involving ancestry-specific allele frequencies (though, we also show how to directly compute individual effects based on their allelic dosages across all the loci, if one has information about the individual's allelic dosages across the loci). We then show that the expected effect under the global model, β^{Glo} , satisfies $\beta^{\text{Glo}} \approx \bar{a} * \beta^{\text{Afr}} + (1 - \bar{a}) * \beta^{\text{Eur}}$. In other words, β^{Glo} is the “best guess” to the true individual effect b if we do not know the individual's specific patterning of local ancestries and allele dosages. This is a necessary approximation because we do not know the exact set of *trans*-modifier variants relative to the focal variant, let alone the allele dosages and epistatic effect sizes. (See also our response to Reviewer 2, Point 3.) Therefore, one could view our approach as indirectly modeling variant effects as a function of the set of all causal loci. We have added a note to the Main Text highlighting this interpretation (**Lines 118-122**).

Future work aligned with what we described in the previous paragraph could explore the use of pathway-conditioned global ancestries, whereby genes or pathways ascertained to modify the effect of a focal variant are directly incorporated into models of variant effect estimation, in case the *trans*-modifier variants themselves cannot be accurately fine-mapped. Examples of works related to this viewpoint include Iakovliev et al., 2023 *AJHG* [PMID: 37164005] and Spiliopoulou et al., 2025 *Arthritis Rheumatol.* [PMID: 39887658].

Re: GxE interaction, we agree on the implication of a linear relationship if global ancestry is capturing such an interaction, which is one reason why we think that the global model is unlikely to fully capture GxE effects (see **Lines 533-542** of Discussion).

4) page 7: Eqs. (9) and (10) assume that effect sizes are estimated without bias, and therefore that there is no confounding, which should be noted.

We have added the following sentence to our manuscript (**Lines 167-169**).

“We assume in our analysis that tagging effects are estimated without bias, though controlling for confounding during estimation is an appreciably challenging task”

5) SI page 32: Under what conditions are the approximations in the derivations of LAACor'Loc and LAACor'Glo valid?

We use $E[X/Y] \approx E[X]/E[Y]$ and $\sqrt{E[X]} \approx E[\sqrt{X}]$ to obtain *approximate expressions* for LAACor'Loc (a conceptual quantity). We were unable to obtain error bounds to this approximation; the closest attempt we made was to apply the more precise approximation $E[X/Y] \approx E[X]/E[Y] - \text{cov}(X,Y)/(E[Y])^2 + \text{var}(X)E[X]/(E[Y])^3$, but we were unable to estimate the covariance quantity up to the relevant order of magnitude for asymptotic analyses. (Concretely, $E[Y] \sim \Theta(p)$, while independence of markers implies $\text{var}(X) \sim O(p)$ and $E[X] \sim O(p)$, ensuring $\text{var}(X)E[X]/(E[Y])^3 \sim o(p)$. But we could not rigorously show $\text{cov}(X,Y) \sim O(p)$, which would suffice to obtain $E[X/Y] \approx E[X]/E[Y]$, since the remaining terms would be $o(p)$.) Regardless, we simulated effect sizes using independent markers from real genotype data, and confirmed our approximate expressions to be very close to the Monte Carlo estimate. This suggests that the asymptotic relationships described in the previous sentence are likely true under the condition of independent markers. No other assumptions are made about allele frequencies, unlike in Proposition 4.2.

6) page 11: For the conditions under which Proposition 4.2 holds, we are directed to Supplementary Section S5 (page 20 of the SI, bullets (A)-(E)). However, some of these conditions are very stringent (for example, equal within-ancestry allele frequencies across markers, identical ancestry assignment across haplotypes in a diploid individual), and therefore should be included explicitly in the statement of the Proposition.

We have included these conditions (see **Lines 373-380** under Proposition 4.2). We agree that these conditions used to derive the analytical results are stringent but we note that in practice the results are fairly robust to violations.

7) page 12: In the caption of Figure 2, it says that "approximate 95% confidence intervals are shown" for the genome-wide average effect correlations displayed, but I cannot see these confidence intervals in the figure.

The confidence intervals are very small and thus obscured by the points. We have shrunk the point estimates and increased the bar lengths to improve the visibility of the bars.

TYPOS/WORDING

8) page 4: "whose haplotypes are columns j and $j+p$ " -- In what sense are these haplotypes? The columns give the two haploid genotypes at causal locus j in the sampled individuals.

We agree that our phrasing here is confusing, so we have removed the clause entirely (refer to **Line 81** for removal).

9) SI page 2: dipoid  diploid

We have fixed this.

Reviewer 2

Overall, I think this paper makes an important theoretical contribution to an area of research that has become especially salient for understanding and improving polygenic score portability. It provides a biologically realistic framework for thinking about causal effect differences across populations, which is something that has been missing from the literature thus far.

Thank you for valuing our work.

Below I outline some specific questions, along with some suggestions for clarifying the models and assumptions in the main text.

1) If phenotypes and effect sizes are both being simulated, I'm unclear on why the GWAS summary statistics are needed for simulation study 2. Can causal and tag variant pairs not simply be simulated by sampling from all pairs of tightly linked variants? What benefit does it provide to condition on GWAS hits?

The motivation for using GWAS hits was to capture the allele frequency distribution and joint distribution of causal and tagging variants included in polygenic scores in practice. We were specifically concerned about systematic differences between GWAS hits and GWAS null variants (e.g., would clumping and thresholding systematically avoid variants whose allele frequencies tend to lie within a certain range, and if so, this would render such variants underrepresented in, or operationally irrelevant to, realistically constructed polygenic scores). We agree that other simulation strategies are possible but also come with caveats — for example, our approach means that we do not need to model the GWAS discovery process. We've added a brief justification to **Lines 274-278**.

2) Similarly, I'm curious about the choice to approximately estimate r^2 and the number of causal variants through a proxy analysis of polygenic score performance. Is there a reason why published heritability or polygenicity estimates wouldn't be suitable for these simulations? By comparison, the approach used in this paper seems more challenging to perform and harder to justify.

Part of the motivation here is again to avoid modelling the GWAS discovery process. Published heritabilities are much higher than PGS r^2 , which is really the quantity that we are trying to capture. There is an alternative strategy where we model all causal variants and the GWAS discovery process, but we think that modelling only the variants included in the PGS (conditioned on the properties of those variants as in the previous point) is simpler to implement and requires fewer assumptions.

3) A few questions on notation in Supplement S2: is it correct to say that b is the true individual-level effect and β is the group-averaged marginal effect? Also, are the betas in this section corresponding to causal betas? (They are missing the apostrophe used to denote causal variant effects elsewhere in the manuscript.)

Yes, b is the true individual-level effect, while β is the group-averaged marginal effect. Yes, the betas in Supplement S2 correspond to causal betas. Thank you for catching this typo. We have added the appropriate apostrophes to Supplement S2.

A note on the possible confusion about β_i^{Glo} being simultaneously a “group-averaged” effect and an individual effect size. We showed that β_i^{Glo} is the “best guess” to the true individual effect b_i , if we do not know their distribution of local ancestries and allele dosages (see response to Reviewer 1, Point 3). This is a necessary assumption if we are unable to provide the exact set of *trans*-modifying variants that exert multiplicative epistatic effects on top of the focal variant effect. If we knew the complete genetic architecture (including all interactions), then we would have, for each causal variant, an accompanying set of *trans*-modifying variants. We would then compute individual-level genetic effects b_i — *without the need for local ancestry information* — as described in Supplement S2.

4) Under the global model, does LAACor depend on the distribution of global ancestry? It's hard to tell from the expressions in Supplement S8, but intuitively, it seems that it could, and it would be interesting to know if this is the case.

Under the global model, it depends on just the mean global ancestry of each group of individuals sharing the same local ancestry, since the global model assigns individual effect sizes based on individual global ancestries. (Averaging individual effect sizes reduces to averaging global ancestries, conditional on having the ancestry-specific effect sizes β_i^{Afr} and β_i^{Eur}). But that is a very small effect, precisely because local ancestry is a poor predictor of global ancestry. See also our response to Reviewer 1, Point 5, which asks for an intuitive explanation about a result regarding LAACor' and LAACor under the global model.

5) It might be nice to give some more intuition for the comparison of causal variant PGS and tag variant PGS - it was difficult to think about why causal variant TotPGS is identical for local and global models, but tag variant TotPGS can distinguish between them.

Thank you for this suggestion that makes our results more accessible and intuitive. We have provided an explanation in **Lines 449-455**.

6) It seems that there are parameter regimes where both ParPGS and TotPGS can distinguish local and global models, and so from Figures 3C and S4-15, it was not obvious to me why the ratio of tagging variant ParPGS and TotPGS could not distinguish local and global models. A tagging variant version of 3A might be nice to include in the supplement, or else the authors could provide some more intuition for the reader.

We agree that this point was not very clearly articulated. To provide some more intuition for the reader, we have included an explanation in **Lines 478-486**.

7) In Figure 3B, is there no uncertainty in the correlation estimated from the different seeds? I was confused by the orientation of the standard error bars.

The uncertainty is very small, and so the standard error bars (oriented like \perp rather than $|-|$) are difficult to visualize. We have noted this in the Main Text (**Lines 423-424**) and have reported the range of standard errors in the figure caption, to reinforce how small the errors are.

8) I found it somewhat difficult to interpret the various betas on a first pass through the manuscript. In section 2.1, it could be helpful to define the ancestry-specific betas before writing out the distribution. Otherwise, I think it is only stated explicitly in Supplement S2 that the ancestry-specific betas are the average causal effect in a population that is "homogeneous" for that particular ancestry. It may potentially be helpful to also include some of the explicit generative models from Supplement S2 in the main text to make the global and local models more transparent to readers.

We have included a sentence introducing the ancestry-specific betas as average genetic effects (**Line 100-102**). We have now provided a verbal description with a reference to the structural causal diagram of the generative model right after the definition of β_i^{Glo} , and articulated the underlying logic of the global model formula as an approximation in the absence of local ancestry and allele dosage information across the *trans*-modifying loci (**Lines 118-122**). See also our detailed response to Reviewer 1, Point 3.

9) The paper uses the phrase "gene-ancestry interaction" in a few places, and I'm not always sure what that means. I assume interaction is being used in the statistical sense, but it's a bit confusing because the paper also talks about interactions in the biological

sense. I also think the nomenclature "gene-ancestry interaction" verges somewhat closer to implying causal effects differ discretely between ancestries, which one wants to avoid. (It's especially notable because in other aspects, the paper does a very good job of emphasizing individual-level variation in causal effects and de-emphasizing continental ancestries as discrete groups.)

We agree that it is important to emphasize that our work does not prove that causal effects differ discretely between ancestries, let alone justify any ill-motivated conclusions drawn from the latter statement. We have added sentences (**Lines 530-533**) to clarify this point. We have removed one instance of "gene-ancestry interaction", and replaced "biologically relevant gene-ancestry interaction" with "statistical gene-ancestry interaction" in the only other place it was used in the Main Text.

10) Figure 1D was somewhat confusing. While I think I understand what the authors are trying to convey, I don't know what specifically is being plotted in the global model diagram and the associated inset.

Figure 1D shows the distribution of individual effect sizes at a particular SNP. The two density plots show this distribution, under the global and the local model. The first density plot conveys two key points: (1) global model implies continuous variation (i.e., high variability) in individual effect size [shown by the density spread]; and (2) local ancestry average effect sizes are highly similar [shown by the closeness of vertical dotted lines]. We have reworded the figure caption so that these two points are stated clearly.

11) If I understood correctly, LAACor is defined as the correlation within an admixed population, and it would be good to make that explicit in 4.1. In the main text, I think this is only stated implicitly via reference to Hou et al.

We have added "within an admixed population" at **Line 310**.

12) The note on demeaning genotypes in Supplement S1 is quite nice, I hadn't thought of that.

Thank you for the positive comment. As suggested by the editor, we have added a more extensive discussion and an explicit pointer to Supplement S1 in the main text (**Lines 95-99**).

13) It would be helpful to reiterate at the start of 4.2 that under this framework, PGS are constructed using betas estimated from a population with only European ancestry (this is explained earlier, in 2.3, but easy to lose track of by the time one gets to 4.2).

We have added the following sentence to the Main Text (**Lines 362-363**).

“We recall that all polygenic scores are constructed using effect sizes estimated from a population with only European ancestry.”

14) I frankly thought the authors were selling themselves short with the sentence in the Discussion "One limitation of our study is that our approach is indirect, relying on a quantitative comparison to other work". Theoretical work can so often help us understand empirical results, and that is true of this paper as well.

Thank you for finding our theoretical approach valuable. On reflection we agree, so we have removed this sentence.

September 18, 2025
RE: GENETICS-2025-308541

Dr. Alan J Aw
University of Pennsylvania Perelman School of Medicine
Genetics
405A Clinical Research Building
415 Curie Boulevard
Philadelphia, Pennsylvania 19104

Dear Dr. Aw:

Congratulations, your manuscript titled "Hidden structure in polygenic scores and the challenge of disentangling ancestry interactions in admixed populations" is accepted for publication in GENETICS! Many thanks for submitting your research to the journal.

Reviewer #2 had a few suggestions for improving the manuscript that you may want to consider. You can view their comments at the bottom of this email.

To Proceed to Publication:

1. Format your article according to GENETICS style: <https://academic.oup.com/genetics/pages/author-guidelines>
2. Ensure that you comply with data and community resource citation guidelines: <https://academic.oup.com/genetics/pages/author-guidelines#section-5-9-2>
3. Upload your final files at <https://genetics.msubmit.net>
4. Add oupsupport@scipris.com and genetics.oup@novatechset.com (or the domains @scipris.com and @novatechset.com) to your email program's "safe senders" list. You will be contacted by both at various points during the production process.

Notes:

- Your currently-accepted manuscript (unedited, as submitted, reviewed, and accepted) will be published at GENETICS and deposited into PubMed as an Advance Access article. Notify sourcefiles@thegsajournals.org before signing your license if you do not wish to publish your article via Advance Access.
- We invite you to submit an original color figure related to your paper for consideration as cover art. Please email your submission to the editorial office or upload it with your final files. You can submit a small-sized image for evaluation, and if selected, the final image must be a TIFF file 2513px wide by 3263px high (8.375 by 10.875 inches; resolution of 600ppi). Please avoid graphs and small type.
- After files are sent to Oxford University Press we use SciPris to manage article licensing and payment. If you do not have a SciPris account, you will receive an email from no-reply@scipris.com to sign up to use Oxford University Press' author portal. After logging in, follow the online instructions to sign your license and arrange any payment due.

If you have any questions or encounter any problems while uploading your accepted manuscript files, please email the editorial office at sourcefiles@thegsajournals.org.

Sincerely,

Hakhamanesh Mostafavi
Associate Editor
GENETICS

Approved by:
Anthony Long
Senior Editor
GENETICS

Review comments (if applicable):

Reviewer #1 :

The authors have comprehensively addressed my previous comments in their revised manuscript.

Reviewer #2 :

I think the paper has been strengthened by the authors' revisions, and my comments have all been sufficiently addressed. I especially appreciate the additions to the Discussion.

I have two very minor comments (that reference line numbers from the tracked changes draft). First, some of the newer language could be revised for clarity. Lines 341-352 are a bit hard to parse on a first read due to sentence length, and the assumptions outlined in Prop. 4.2 are at times vague (e.g. (A) seems to imply all markers have the same frequency, and I do not think this is what the authors meant?). Second, it might be worth citing [13] in line 536.